# An alternative splicing switch shapes neurexin repertoires in principal neurons versus interneurons in the mouse hippocampus

Thi-Minh Nguyen, Dietmar Schreiner[†], Le Xiao[†], Lisa Traunmüller, Caroline Bornmann, Peter Scheiffele*

Biozentrum, University of Basel, Basel, Switzerland

**Abstract** The unique anatomical and functional features of principal and interneuron populations are critical for the appropriate function of neuronal circuits. Cell type-specific properties are encoded by selective gene expression programs that shape molecular repertoires and synaptic protein complexes. However, the nature of such programs, particularly for post-transcriptional regulation at the level of alternative splicing is only beginning to emerge. We here demonstrate that transcripts encoding the synaptic adhesion molecules neurexin-1,2,3 are commonly expressed in principal cells and interneurons of the mouse hippocampus but undergo highly differential, cell type-specific alternative splicing. Principal cell-specific neurexin splice isoforms depend on the RNA-binding protein Slm2. By contrast, most parvalbumin-positive (PV$^+$) interneurons lack Slm2, express a different neurexin splice isoform and co-express the corresponding splice isoform-specific neurexin ligand Cbln4. Conditional ablation of *Nrxn* alternative splice insertions selectively in PV$^+$ cells results in elevated hippocampal network activity and impairment in a learning task. Thus, PV-cell-specific alternative splicing of neurexins is critical for neuronal circuit function

*For correspondence: peter.
scheiffele@unibas.ch

[†]These authors contributed
equally to this work

**Competing interests:** The
authors declare that no
competing interests exist.

**Reviewing editor:** Sacha B
Nelson, Brandeis University,
United States

## Introduction

Specific synaptic connectivity and function are essential for the appropriate operation of neuronal circuits. A large degree of this structural and functional specificity is thought to be genetically encoded. For example, synaptic partners express matching pairs of adhesive factors or afferents are repelled from inappropriate targets through chemorepulsive signaling molecules (*Sanes and Yamagata, 2009*; *Shen and Scheiffele, 2010*). Gene families encoding large numbers of isoforms generated through multiple genes, alternative promoters and extensive alternative splicing, hold the potential to generate recognition tags for specific trans-synaptic interactions (*Baudouin and Scheiffele, 2010*; *Reissner et al., 2013*; *Takahashi and Craig, 2013*; *Schreiner et al., 2014b*; *Li et al., 2015*). However, given the difficulty of mapping endogenous splice isoform repertoires it is poorly understood how splice isoforms are differentially distributed across neuronal cell types.

Neurexins (*Nrxn1,2,3*) represent one gene family of highly diversified synaptic adhesion molecules. Through the use of alternative promoters (alpha and beta) and alternative splicing at up to six alternatively spliced segments (AS1-6) more than 1300 transcripts are generated that are expressed in the mature mouse nervous system (*Schreiner et al., 2014a*; *Treutlein et al., 2014*; *Schreiner et al., 2015*). Isoform diversity scales with the cellular complexity of brain regions and one purified cell population was shown to be strongly enriched for a subset of isoforms (*Schreiner et al., 2014a*). This indicated that at least some neurexin isoforms are enriched in a cell type-specific manner. Pairwise comparisons of relative transcript levels recovered from single cells suggested that

individual cells within one cell type might exhibit more similar alternative exon usage than cells from divergent origins (*Fuccillo et al., 2015*). However, the actual alternative exon incorporation rates in interneuron and principal neuron populations have not been examined.

Importantly, individual splice insertions in the neurexin proteins control biochemical interactions with an array of synaptic ligands (*Baudouin and Scheiffele, 2010*; *Reissner et al., 2013*). Based on ectopic expression and knock-out experiments in mice it has been postulated that neurexin isoforms might contribute to an alternative splice code for selective synaptic interactions and differ in their tethering at neuronal synapses (*Boucard et al., 2005*; *Chih et al., 2006*; *Graf et al., 2006*; *Fu and Huang, 2010*; *Futai et al., 2013*; *Aoto et al., 2015*; *Traunmüller et al., 2016*). However, interpretation of such findings is complicated by the fact that the manipulations of isoforms were conducted in cells where endogenous isoform repertoires were unknown. In the human population mutations in *Nrxn1,2*, and 3 are associated with neurodevelopmental disorders such as autism and schizophrenia (*Kim et al., 2008*; *Yan et al., 2008*; *Kirov et al., 2009*; *Rujescu et al., 2009*; *Gauthier et al., 2011*; *Vaags et al., 2012*). Global deletion of the majority of *Nrxn1* or *Nrxn3* transcripts in mice or global perturbation of the *Nrxn* alternative splicing at AS4 disrupts function and plasticity of glutamatergic and GABAergic synapses (*Missler et al., 2003*; *Etherton et al., 2009*; *Aoto et al., 2013*; *Traunmüller et al., 2016*). However, the function of neurexin isoforms in interneurons has not been examined with targeted approaches.

In this study we uncover a major alternative splice isoform switch that distinguishes glutamatergic and GABAergic cell populations in the hippocampus. We demonstrate that *Nrxn1,2,3α* transcripts are commonly expressed in pyramidal cells and fast-spiking GABAergic interneurons expressing the calcium binding protein parvalbumin (PV$^+$ cells). However, pyramidal and PV$^+$ cells exhibit highly differential incorporation rates of alternative exons at AS4. This alternative splicing switch depends on the differential expression of RNA-binding proteins and coincides with the cell type specific expression of a neurexin splice isoform-specific ligand. Selective disruption of PV$^+$ cell splice variants in mice results in functional and behavioral abnormalities. Thus, interneuron-specific alternative splicing of neurexins is important for normal circuit function.

## Results

### Neurexin alpha mRNAs are highly expressed in pyramidal cells and PV$^+$ interneurons of the mouse hippocampus

To begin to assess the differential expression and functional relevance of neurexin isoforms in mouse neuron populations, we first examined the six primary *Nrxn1,2,3α/β* transcripts by in situ *hybridization*. Similar to what has been reported for rat hippocampus (*Ullrich et al., 1995*) we find significant expression of *Nrxn1,2,3α* transcripts in *cornus ammonis* (CA) pyramidal cells as well as presumptive interneurons (*Figure 1—figure supplement 1A and B*). To specifically interrogate *Nrxn* transcripts in genetically defined cell populations we tagged ribosomes in CA pyramidal cells and PV$^+$ interneurons, a population of GABAergic, fast-spiking cells that encompasses chandelier and basket cells (*Hu et al., 2014*). We used a conditional HA-tagged Rpl22 allele (*Sanz et al., 2009*) crossed with *CamK2$^{cre}$* (*Tsien et al., 1996*) and *Pvalb$^{cre}$* drivers (*Hippenmeyer et al., 2005*), respectively (see *Figure 1* and also *Figure 1—figure supplement 2* for the selectivity of Rpl22-HA expression in the resulting CamK2$^{Ribo}$ and PV$^{Ribo}$ mice). RiboTrap purifications (*Heiman et al., 2014*) of polysome-associated mRNAs from adolescent (P24-P28) CamK2$^{Ribo}$ or PV$^{Ribo}$ mice yielded enrichment of mRNAs from the respective cell populations as confirmed by real-time quantitative PCR (qPCR). Thus, CamK2$^{Ribo}$ preparations showed enrichment of C*amK2* mRNA and the CA1-specific marker *Wsf1* (*Wolfram syndrome 1*) (*Figure 1B*) which is consistent with high cre-activity in the CA1 area (*Figure 1A* and *Figure 1—figure supplement 2A*) and a de-enrichment of interneuron and astrocyte markers. By contrast, PV$^{Ribo}$ preparations were highly enriched in the common interneuron marker (*Gad1*) and in the mRNAs specifically expressed in fast-spiking basket cells (*Pvalb, Erbb4*) (*Figure 1B* and *Figure 1—figure supplement 1B*). *Nrxn1,2,3* mRNAs were recovered in both CamK2$^{Ribo}$ and PV$^{Ribo}$ cell-derived transcript preparations (note that *Nrxn3β* expression in mouse hippocampus is low and could not be reliably detected – see *Figure 1—figure supplement 1A–C*). Notably, amongst all neurexin transcripts *Nrxn3α* was most highly enriched in the PV-cell population (*Figure 1C*). PV-cell expression of *Nrxn3α* was further confirmed by dual labeling with in situ

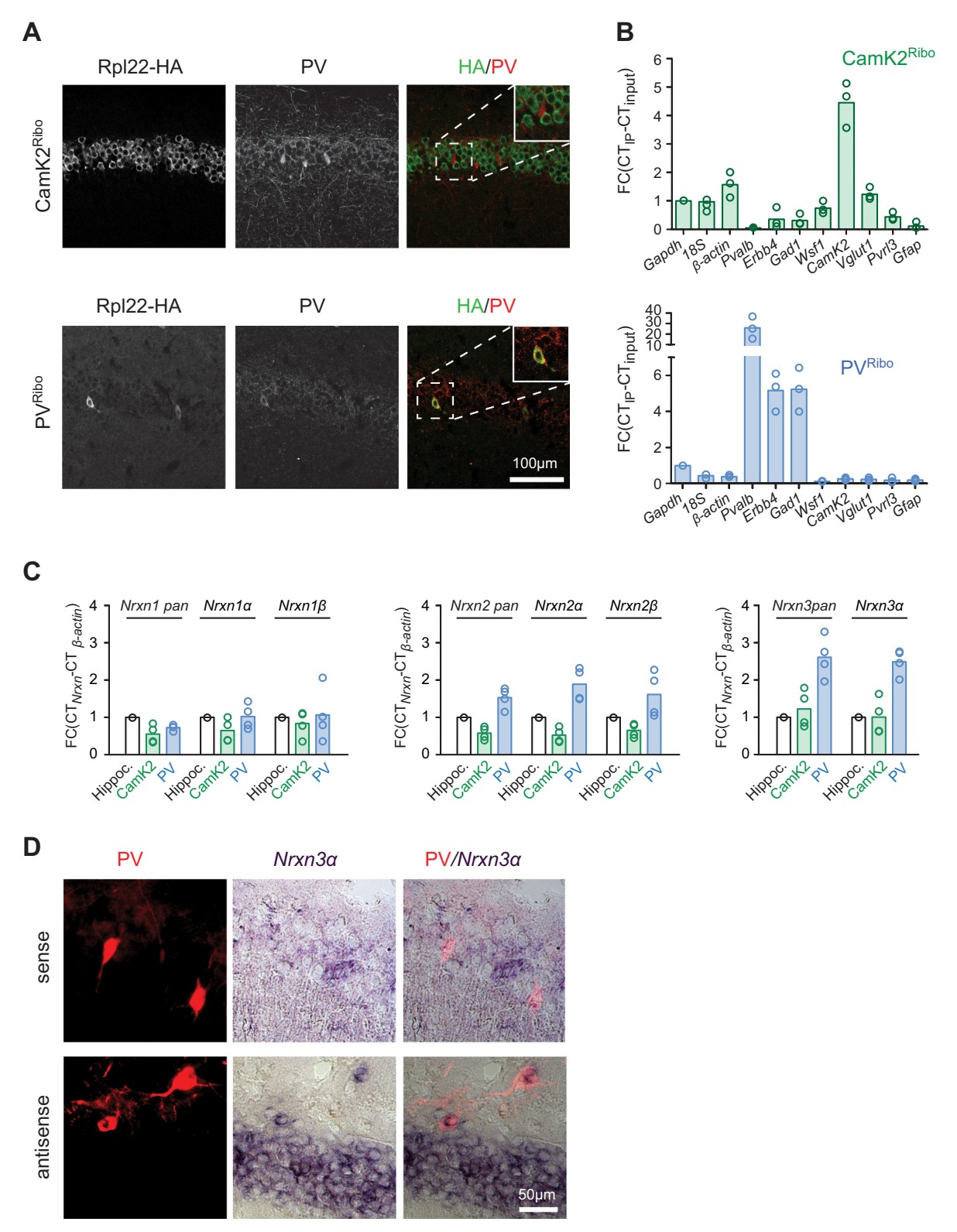

**Figure 1.** Neurexin transcripts are highly expressed in PV⁺ interneurons. (A) Conditional expression of Rpl22-HA in pyramidal cells (CamK2^Ribo) and fast-spiking interneurons (PV^Ribo) in mouse hippocampus (postnatal day 28). Immunoreactivity for epitope-tagged Rpl22 (anti-HA, green in merge), anti-PV (red in merge) is shown. Inset in merge shows an enlargement of the boxed area. (B) Purification of cell type-specific polysome-associated transcripts by RiboTrap affinity purification. Transcript levels were assessed by real-time qPCR for general markers of glutamatergic neurons (*Vglut1*, *CamK2*), markers

*Figure 1 continued on next page*

*Figure 1 continued*
of CA1 pyramidal cells (*Wsf1*), CA3 (*Pvrl3*), common GABAergic marker (*Gad1*), and fast-spiking markers (*Pvalb* and *Erbb4*). Enrichment in the immunoisolate (IP) was calculated relative to the input and was normalized to *Gapdh* (n = 3 independent mRNA preparations). (C) Expression of *Nrxn1,2,3* transcripts in PV⁺ and CamK2⁺ cells was examined by real-time qPCR. Transcript levels in each preparation were normalized to the level of *β-actin* transcripts and enrichment in the immunoisolate (IP) was calculated relative to the input levels in total hippocampus (n = 4 independent mRNA preparations). Neurexin three beta transcripts were not reliably detectable with our assays in the hippocampus due to low expression (see *Figure 1—figure supplement 1C* for further information). (D) Expression of *Nrxn3α* in PV⁺ cells in CA1 (postnatal day 21) revealed by dual labeling with in situ *hybridization* using *Nrxn3α* probes and immunostaining using antibody against RFP in mice where PV⁺ cells are genetically marked by cre-dependent expression of red fluorescent protein (*Pvalb^cre^::Ai9^Tom^*).

The following figure supplements are available for figure 1:

**Figure supplement 1.** Detection of primary neurexin transcripts by in situ *hybridization*.
**Figure supplement 2.** Conditional Rpl22-HA expression in mouse hippocampus.

*hybridization* using *Nrxn3α* probes and immunostaining in mice where PV⁺ cells were genetically labelled with red fluorescent protein (*Pvalb^cre^::Ai9^Tom^*) (*Figure 1D*).

## Differential alternative splicing and RNA binding protein expression in principal versus PV⁺ interneurons

To quantitatively probe alternative exon incorporation rates in the *Nrxn* transcripts we used radioactive PCR amplification with primers flanking the alternatively spliced segments (AS2-AS6). Importantly, this method is not plagued by problems of differential PCR primer efficiencies that are encountered in isoform-specific real-time qPCR. We uncovered similar usage of alternative exons at AS3 across all preparations. Interestingly, *Nrxn1 AS6* and *Nrxn2 AS2* exhibited differential usage in PV− versus CamK2 cells. Moreover, for all three *Nrxn* transcripts (*Nrxn1,2,3*) the alternative exon incorporation rates at AS4 were remarkably divergent between the two cell populations (*Figure 2B and C* and *Figure 2—figure supplement 1A*). While in CamK2^Ribo^ cells the cassette exon at AS4 was largely skipped there was a high level of alternative exon inclusion in PV^Ribo^ cells. Thus, highly selective, cell type-specific alternative exon incorporation rates of *Nrxn* mRNAs generate divergent *Nrxn* splice isoform repertoires in glutamatergic CA pyramidal cells and PV⁺ interneurons.

Neurexin alternative splicing at AS4 is regulated by the STAR-family of RNA-binding proteins, in particular the protein Slm2 which regulates skipping of the alternative exon (*Iijima et al., 2011*; *Ehrmann et al., 2013*; *Iijima et al., 2014*; *Traunmüller et al., 2014*, *2016*). Thus, we tested whether cell type-specific alternative splicing at AS4 may be a consequence of differential expression of Slm2 in pyramidal versus PV⁺ cells that are targeted in the *CamK2^cre^* and *Pvalb^cre^* transgenic lines, respectively. As reported previously (*Stoss et al., 2004*; *Iijima et al., 2014*), we observed high expression of Slm2 protein in pyramidal cells, and more than 90% of CamK2^Ribo^-positive cells were Slm2-positive (*Figure 2E*). By contrast, 75% of cells marked in *Pvalb^cre^::Ai9^Tom^* mice in hippocampus area CA1 do not express Slm2 (*Figure 2E*). In particular, cells within or close to the pyramidal cell layer were Slm2-negative (*Figure 2D and E* - note that the percentage of Slm2-expressing 'PV⁺ cells' depends on the anatomical position within the hippocampus and the use of genetic versus antibody-labeling to define 'PV⁺ cells', *Figure 2—figure supplement 1D and E*). We further explored differential expression of STAR-family RNA binding proteins by real-time qPCR on polysome-associated transcripts from CamK2^Ribo^ and PV^Ribo^ mice (*Figure 2F*). Consistent with the immunohistochemistry results, we observed an enrichment of *Slm2* transcripts in the CamK2^Ribo^ population and a de-enrichment in the PV^Ribo^ preparations (similarly the *Slm2* paralogues *Slm1* and *Sam68* were de-enriched). Given that global ablation of Slm2 results in a significant loss of the skipped (AS4-) neurexin isoforms (*Ehrmann et al., 2013*; *Traunmüller et al., 2014*), this strongly suggests that differential expression of Slm2 in pyramidal versus PV⁺ cells drives the cell type-specific skipping of alternative exons at AS4 in pyramidal cells of the hippocampus. Finally, we observed that ectopic expression of Slm2 in cells that normally do not express high levels of STAR proteins (cerebellar granule cells in culture) was sufficient to shift alternative splicing in favor of *Nrxn AS4-* isoforms (*Figure 2—figure supplement 2*).

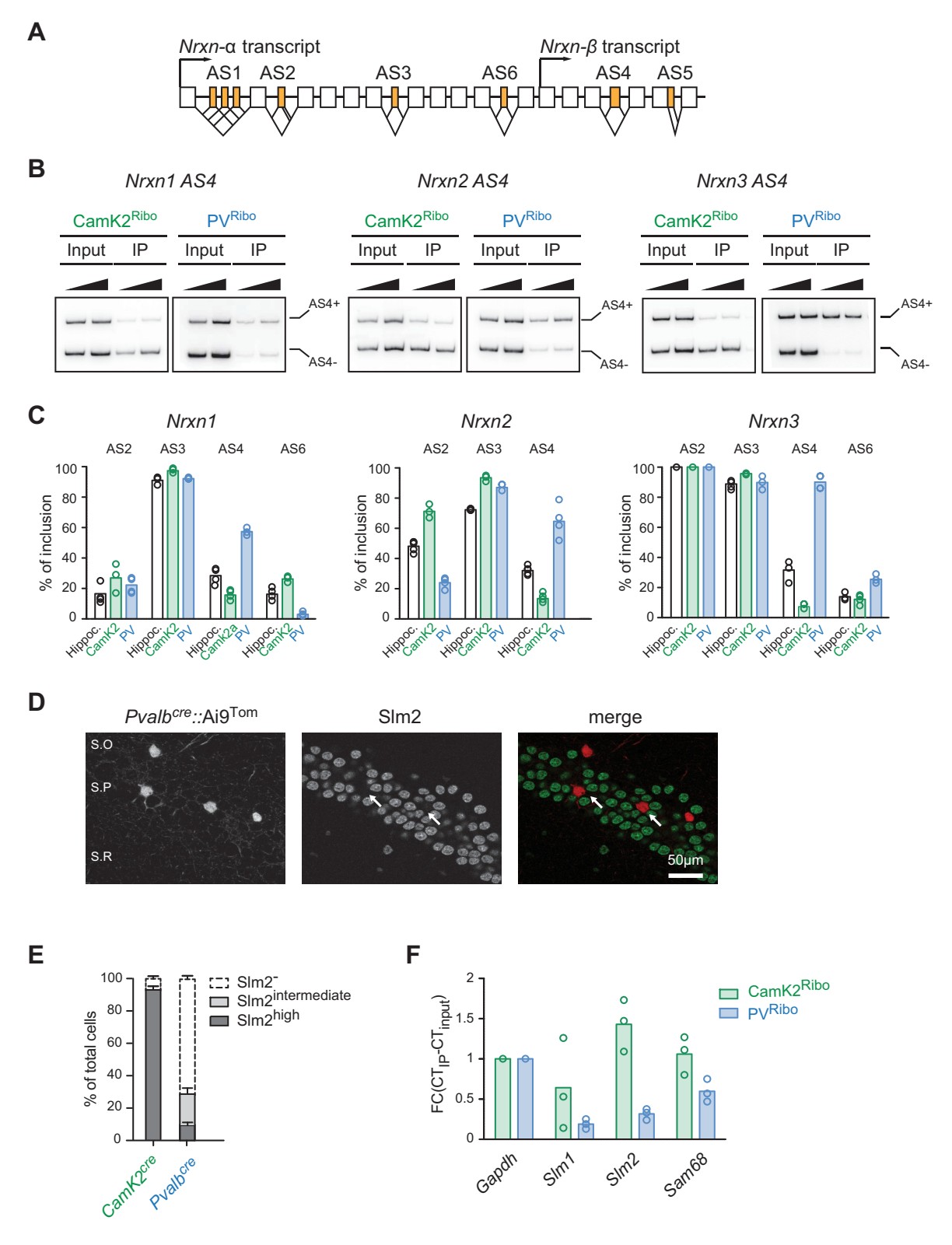

**Figure 2.** Cell type-specific alternative splicing. (**A**) Schematic illustrating exon-intron structure of the *Nrxn* gene (example based on mouse *Nrxn1*). Alternatively spliced segments are numbered (AS1-6) and alternative exons are highlighted in orange, constitutive exons in white. (**B**) Analysis of alternative splicing pattern in total hippocampus (input), Camk2[Ribo] and PV[Ribo] preparations. Radioactive PCR amplifications for *Nrxn1,2,3 AS4*. For each sample two PCR reactions with increasing cDNA input are shown. (**C**) Quantifications of alternative exon insertion rates at alternatively spliced segments

*Figure 2 continued on next page*

*Figure 2 continued*

(AS) 2,3,4, and six in *Nrxn1,2,3* transcripts for total hippocampus (input), CamK2$^{Ribo}$ and PV$^{Ribo}$ preparations. The insertion rates were measured by radioactive PCR with limiting cycle numbers. Raw data for the radioactive PCR amplifications are shown in *Figure 2—figure supplement 1* (n= 3-4 independent mRNA preparations). (D) Expression of Slm2 in PV$^{+}$ cells in mouse hippocampus (postnatal day 25–30) was examined using *Pvalb$^{cre}$:: Ai9$^{Tom}$* mice. Dual immunohistochemistry on vibratome sections reveals high Slm2 expression in hippocampal pyramidal cells but no detectable expression in the majority of PV$^{+}$ cells in and adjacent to the *stratum pyramidale* (S.O: stratum oriens, S.P: statrum pyramidale, S.R: stratum radiatum). (E) Quantification of the percentage of CamK2$^{Ribo}$ and *Pvalb$^{cre}$::Ai9$^{Tom}$* positive cells that show specific Slm2 immunoreactivity. (n = 5 mice for CamK2$^{Ribo}$ with a total of 2312 cells and five mice for *Pvalb$^{cre}$::Ai9$^{Tom}$* with a total of 244 cells, mean + SEM). (F) mRNA expression of STAR-family RNA-binding proteins was assessed by real-time qPCR in CamK2$^{Ribo}$ and PV$^{Ribo}$ mRNA preparations (n = 3 independent mRNA preparations).

The following figure supplements are available for figure 2:

**Figure supplement 1.** Assessment of alternative exon incorporation rates by radioactive PCR.

**Figure supplement 2.** Ectopic expression of Slm2 is sufficient to drive expression of Nrx AS4- isoforms.

## AS4 + splice insertions selectively enhance the function of neurexins towards GABAergic postsynaptic components

Neurexin AS4+ and AS4- isoforms differ in biochemical interactions with several synaptic ligands. In particular, AS4+ isoforms bind to a class of extracellular ligands called cerebellins (Cbln1-4) (*Uemura et al., 2010*; *Ito-Ishida et al., 2012*). We hypothesized that neurexin alternative splicing and Cbln expression might coincide. We detected an enrichment of *Cbln4*-encoding transcripts in the PV$^{+}$ cell population (*Figure 3A*). By contrast, the CamK2$^{Ribo}$ population was de-enriched for both, *Cbln2* and *Cbln4* transcripts (note that *Cbln1* and *Cbln3* show very low expression in the mouse hippocampus as compared to the cerebellum, *Figure 3—figure supplement 1A*). Thus, pyramidal neurons in the hippocampus express high levels of Slm2 which drives production of AS4- splice isoforms, and there are low levels of Cbln2 and 4. Conversely, the PV$^{+}$ cell population largely lacks STAR-family RNA-binding proteins (Sam68, Slm1, Slm2), exhibit high levels of alternative exon inclusion at AS4, and the cells co-express Cbln4. Despite the similarities between Cbln1,2, and 4 it has remained unclear to what extent Cbln4 interacts with Nrx AS4+ proteins. In fact, binding to Nrx3 AS4+ isoforms has been reported to be low or not detectable (*Joo et al., 2011*; *Matsuda and Yuzaki, 2011*; *Wei et al., 2012*). Using in vitro binding assays with Cbln4 from conditioned media we confirmed binding to cells expressing Nrx1α AS4+ but significantly lower binding to Nrx3 AS4+ isoforms (*Figure 3B–D*). We then tested another assay configuration where Cbln proteins and Nrx isoforms are co-expressed in the same cell (*Figure 3E*). In this assay, Cbln1 or 4 were strongly retained at the cell surface of cells co-expressing Cbln1 or Cbln4 together with Nrx3 AS4+ isoforms but not Nrx3 AS4- (*Figure 3F and G*). Thus, Cbln4 indeed associates not only with Nrx1α but also Nrx3α proteins in a AS4 splice-isoform-specific manner (further studies will be required to confirm whether these interactions are direct or whether they may involve additional linker proteins).

The observation that AS4+ isoforms are highly expressed in PV$^{+}$ cells is notable since AS4+ variants of beta-neurexin exhibit a synaptogenic activity preferentially for GABAergic over glutamatergic postsynaptic structures in cell culture assays (*Chih et al., 2006*; *Graf et al., 2006*). However, the majority of neurexin proteins in vivo are alpha isoforms (*Schreiner et al., 2015*) and the impact of AS4+ insertions on the synaptogenic activity of alpha neurexins has not been examined. Thus, we tested whether the presence or absence of AS4 insertions might modify the synaptogenic activity of Nrxα proteins. Inclusion of the 30 amino acid AS4+ insertion elevated the action of Nrx3α proteins towards GABAergic postsynaptic sites (*Figure 3H and I* and *Figure 3—figure supplement 1C*. Note that for Nrx1α we did not observe the same impact of the AS4+ insertions, *Figure 3—figure supplement 1C–E*). Co-expression of Cbln4 with the Nrx isoforms neither enhanced nor inhibited the synaptogenic activity in this assay (*Figure 3H and I*). Thus, this function either does not directly involve Cbln4 or there is sufficient endogenous Cbln4 expressed in the neuronal cultured to transduce a neurexin signal towards the interaction partners at GABAergic postsynaptic sites.

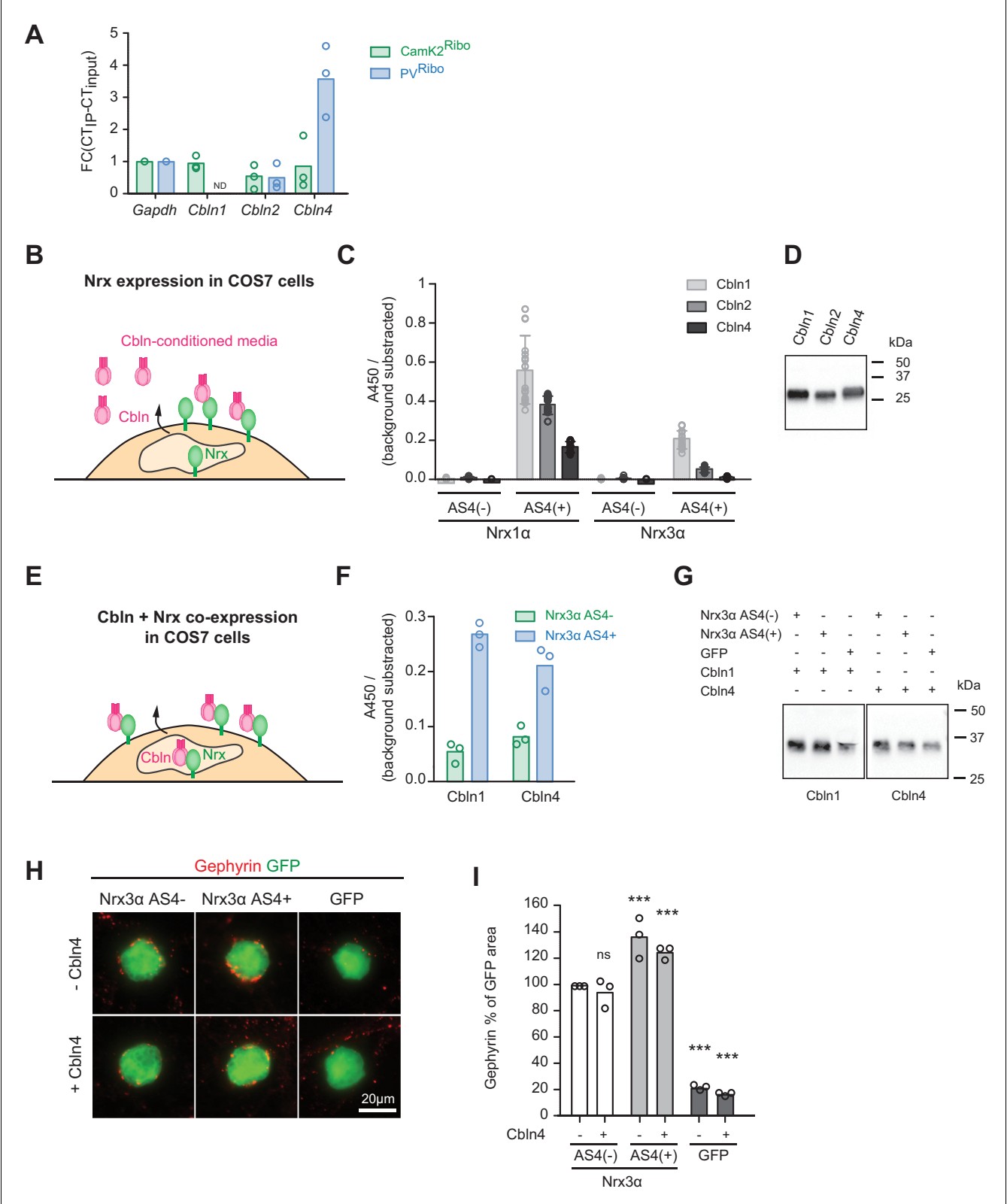

**Figure 3.** Regulation of Nrx3 interaction by AS4- splice insertions. (**A**) *Cbln1, 2, 3* and *4* mRNA expression level was assessed by real-time qPCR in CamK2[Ribo] and PV[Ribo] preparations from mouse hippocampus. Transcript enrichment in the immunoisolate (IP) was calculated relative to the transcript level in the total hippocampus (input) and was normalized to *Gapdh* (n = 3 independent mRNA preparations). Note that *Cbln1* mRNA could not be detected in the PV[Ribo] mRNA preparation. ND: not detected. (**B**) Schematic for experimental setup: binding of Cbln1,2,4 proteins to COS7 cells

*Figure 3 continued on next page*

*Figure 3 continued*

expressing Nrx. Conditioned media containing V5-epitope-tagged Cbln proteins were applied to neurexin expressing cells and binding was determined using a HRP-mediated colorimetric reaction (see methods for details). (C) Quantification of Cbln-Nrx surface binding signals. The background was substrated from the signal. Single dots in the graph represent value of single well measurements (n = 20 measurements per condition, mean ± SD). (D) Expression level of V5-epitope tagged Cbln1, 2 and 4 in conditioned media from COS7 cells was probed by Western blotting analysis with anti-V5 antibodies. (E) Schematic for experimental setup: cell surface accumulation of Cbln1 or Cbln4 that were co-expressed with Nrx3α AS4 with or without splice insert in COS7 cells. The binding was determined using a HRP-mediated colorimetric reaction (see methods for details). (F) Quantification of cell surface accumulation of Cbln-Nrx. Signals were background subtracted and quantified from three independent experiments (n = 3 independent cell cultures). (G) Expression of myc-epitope tagged Cbln1 and 4 was probed by Western blotting. (H) Heterologous cell assays comparing synaptogenic activities of Nrx3 splice variants in presence and absence of Cbln4. HEK293 cells co-expressing Cbln4 with Nrx3α (with or without AS4 insertion) or GFP were introduced into cultures of hippocampal neurons and inhibitory postsynaptic structures were visualized by immunostaining with anti-gephyrin antibodies (red). (I) The density of gephyrin-positive structures, relative to the HEK293 cell area (GFP-positive) was quantified (n = 3 independent cultures,with ≥39 and 20 cells analyzed per condition for Nrx and GFP, respectively, mean ± SEM, Dunnet's multiple comparison test, ***$p < 0.001$, **$p < 0.01$, *$p < 0.05$, [ns] not significant).

The following figure supplement is available for figure 3:

**Figure supplement 1.** Additional assays for Cbln expression and function.

## Selective deletion of PV⁺ cell specific splice variants impairs short-term memory and increases network activity in mice

To assess whether the molecular program identified here is functionally relevant we generated *Nrxn3 AS4* splice isoform-specific conditional knock-out mice in which exon 21 (which encodes the alternative insertion at AS4) is flanked by loxP sites (*Figure 4A*). Upon germline ablation we observed a complete loss of exon 21-containing *Nrxn3* transcripts (*Figure 4B*). Loss of Nrx3 AS4+ protein isoforms was confirmed by serial reaction monitoring mass-spectrometry assays (*Schreiner et al., 2015*) designed to specifically quantify expression of AS4+ isoforms (*Figure 4C*). Importantly, alternative splicing in *Nrxn1* and *2* as well as total Nrx1,2,3 protein levels were unchanged (*Figure 4B and D*). Thus, the *Nrxn3* exon 21 mutation specifically alters the splice isoform identity but not total protein or transcript neurexin levels in the hippocampus.

We then conditionally ablated exon 21 selectively in PV⁺ cells using *Pvalb^cre* mice (*Nrxn3^ex21flox^:: Pvalb^cre^*, in the following referred to as *Nrxn3^ex21ΔPV^*). Considering that neurexin proteins derived from the *Nrxn1,2,3* genes may exhibit overlapping and/or redundant functions, we combined *Nrxn3^ex21ΔPV^* mutants with conditional *Nrxn1^ex21flox^* knock-out mice (*Traunmüller et al., 2016*). The conditional *Nrxn3^ex21ΔPV^* single mutant mice and *Nrxn1/3^ex21ΔPV^* double mutant mice were born at Mendelian ratios and were viable and fertile (*Figure 4—figure supplement 1*). Overall, hippocampal anatomy and density of PV-immunoreactive cells was not detectably changed, indicating that the mutations did not result in severe developmental defects (*Figure 4E*). In the conditional single and double knock-out mice there was no change in the density of perisomatic synapses detected by immunostaining for the postsynaptic GABAergic marker Neuroligin2 (NL2) and synaptotagmin2 (Syt2) (which selectively marks PV⁺ cell terminals [*Sommeijer and Levelt, 2012*]) (*Figure 5A and B* and *Figure 5—figure supplement 1A and B*). Moreover, we did not detect any change in inhibitory synapses formed between PV⁺ interneurons as identified as Syt2-positive structures formed onto PV⁺ somata (*Figure 5—figure supplement 2*).

The ultrastructure of perisomatic PV⁺ interneurons output synapses formed onto pyramidal cells (identified based on location and ultrastructural characteristics [*Takács et al., 2015*]) was unaltered in the conditional single and double knock-out mice (*Figure 5C–F* and *Figure 5—figure supplement 1C–F*). Moreover, frequency and amplitude of miniature inhibitory postsynaptic currents (mIPSCs) recorded from CA1 neurons in conditional single and double knock-out mice were unchanged (*Figure 5G and H* and *Figure 5—figure supplement 1G and H*). These findings indicate that basal inhibitory synaptic transmission in the CA1 pyramidal cells and synaptic structure are not severely affected by the ablation of the PV-cell specific *Nrxn1* and *Nrxn3 AS4+* isoforms.

As a more general read-out for the function of inhibitory networks in the hippocampus, we scored the density of c-fos immunoreactive cells in hippocampal sub-regions DG, CA3, CA1 (*Figure 6A*). These experiments revealed a significant elevation in the density of c-fos⁺ cells in CA3 regions of the

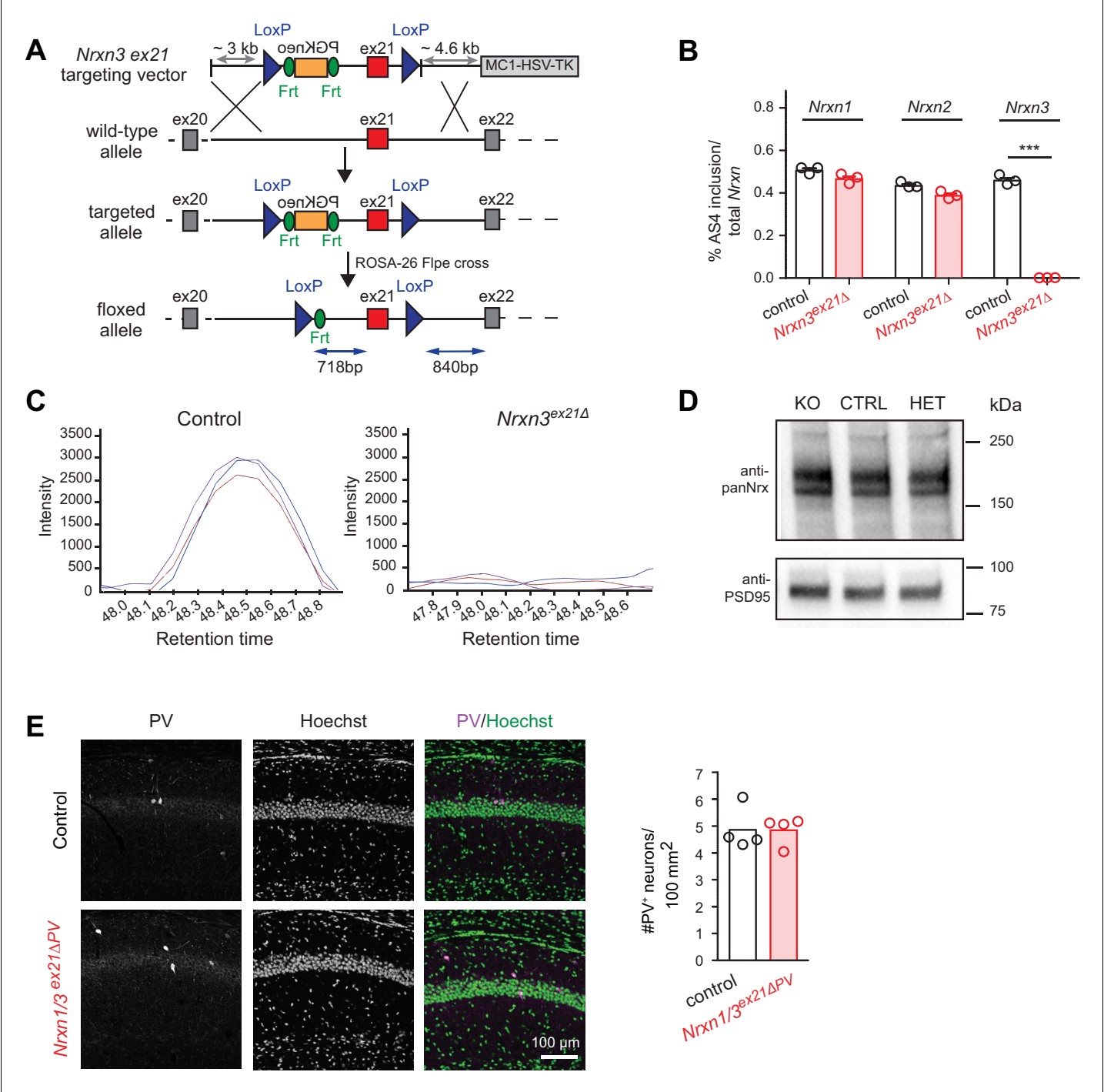

**Figure 4.** Genetic ablation of *Nrxn3* AS4 insertion in mice. (**A**) Targeting strategy for conditional ablation of *Nrxn3* AS4 insertion (*Nrxn3^{ex21ΔPV}* conditional knock-out). See Experimental Procedures for details. (**B**) Analysis of *Nrxn1,2* and *3 AS4* mRNA in control and in *Nrxn3^{ex21Δ}* (n = 3 mice for each genotype, mean+SEM, unpaired t-test p<0.0001). (**C**) Selected Reaction Monitoring MS/MS transitions for an endogenous AS4-specific peptide detected in wild-type and *Nrxn3^{Δex21}* knock-out forebrain extracts. Synaptic proteins were enriched in Triton X-100-insoluble fractions from adult mice. The colored lines show signal intensities for several transitions derived from the same peptide. (**D**) Expression level of Nrx using pan-Nrx antibodies on Triton X-100-insoluble fractions from control (CTRL), *Nrxn3^{ex21Δ}* knock-out (KO) and *Nrxn3^{ex21Δ/+}* heterozygous (HET) mice was determined by Western blotting. (**E**) Immunolabelling of PV⁺ interneurons in the CA1 region of hippocampus of control and and *Nrxn1/3^{ex21ΔPV}* PV conditional double knock-out. PV⁺ interneurons were identified using anti-PV (magenta in overlay) and Hoechst (green in overlay, to counterstain cell nuclei). Quantifications of PV⁺ cell density in CA1 per 100 mm² at postnatal day 24–26. Single dots in the graph represent means of PV⁺ cells per animal (n = 4 mice for each genotype, mean).

*Figure 4 continued on next page*

*Figure 4 continued*

The following figure supplement is available for figure 4:

**Figure supplement 1.** Analysis of weight and Mendelian frequencies in *Nrxn3^{ex21ΔPV}* and *Nrxn1/3^{ex21ΔPV}* conditional knock-out mice.

*Nrxn1/3^{ex21ΔPV}* mice (*Figure 6B*), indicating that network activity is elevated upon ablation of the PV-cell-specific *Nrxn* splice insertions. Given these alterations in hippocampal network activity, we tested whether *Nrxn1/3^{ex21ΔPV}* mice display behavioral deficits using a novel object recognition task which – at least in part – relies on hippocampal function (*Cohen and Stackman, 2015*). In this task, control mice exhibit a significant preference for exploring novel over familiar objects. By contrast, we observed that *Nrxn1/3^{ex21ΔPV}* mutant mice spent equal time exploring familiar and novel objects, indicating a defect in short-term memory (*Figure 6C and D*). These defects were not a consequence of general behavioral disruptions as the mice showed similar total object exploration times, normal locomotion, and no signs of anxiety-related behaviors (*Figure 6E–K*). Thus, PV-cell-specific loss of *Nrxn1* and *3 AS4+* splice variants results in impairment in neuronal circuit function.

## Discussion

In this study we identify an alternative splicing switch that distinguishes principal and PV$^+$ interneurons in the hippocampus. First, we demonstrate that alpha neurexin transcripts are commonly expressed in pyramidal cells as well as PV$^+$ interneurons. We then show that alternative splicing at the alternatively spliced segment four (AS4) is a major driver for the generation of divergent neurexin isoforms in PV$^+$ cells and pyramidal cells. Differential alternative splicing emerges from differential expression of the RNA-binding protein Slm2 that drives selective production of AS4- splice isoforms in pyramidal cells. By contrast, Slm2 is absent from the majority of PV$^+$ cells which produce AS4+ isoforms and co-express the splice isoform-specific ligand Cbln4. Finally, we provide evidence that selective disruption of the PV$^+$ cell-associated AS4+ isoforms results in increased hippocampal network activity and impairs short-term memory, demonstrating that the cell type-specific alternative splice variants are indeed relevant for circuit function.

### Cell-type specific alternative splicing machinery instructs the neurexin isoform code

Previous work provided evidence for cell type-specific repertoires of full-length neurexin transcripts (*Schreiner et al., 2014a*). Pairwise comparison of alternative exon amplification indicated similar exon amplification from single cells within one class of cells (*Fuccillo et al., 2015*). However, these approaches did not provide actual exon incorporation rates across cell types which are a prerequisite for understanding the logic of cell type-specific function of endogenous isoforms.

Our observation that AS4 insertion-containing *Nrxn* mRNAs are dominant in PV$^+$ cells and abundant in the entire hippocampus was surprising as it had been previously suggested that these *Nrxn3* AS4+ isoforms make up less than 10% of *Nrxn3* in the mouse hippocampus (*Aoto et al., 2013*). We note that the estimates of exon incorporation rates based on extrapolation from several independent qPCR assays can make it difficult to reliably assess isoform contents due to differential primer efficiencies. The high abundance of AS4+ isoforms in the hippocampus reported in the present study is supported by radioactive and semi-quantitative PCR assays, as well as previous mass-spectrometric assays that probe Nrx3 AS4+ variants on the protein level (*Figure 2* and [*Ehrmann et al., 2013*; *Traunmüller et al., 2014*; *Schreiner et al., 2015*]). Thus, we conclude that these variants are indeed significantly expressed. More importantly, we demonstrate that they are specifically enriched in interneurons but largely absent from pyramidal cells in the hippocampus.

The highly selective alternative splicing choices in the neurexin pre-mRNAs suggest that they result from neuronal cell-type specific expression of alternative splicing factors. The high expression of *Nrxn1,2,3 AS4-* variants in pyramidal cells discovered here correlates with the high expression of Slm2 protein in pyramidal cells and the near complete loss of these variants in Slm2 knock-out mice (*Ehrmann et al., 2013*; *Traunmüller et al., 2014*). Moreover, we demonstrate that Slm2 is absent

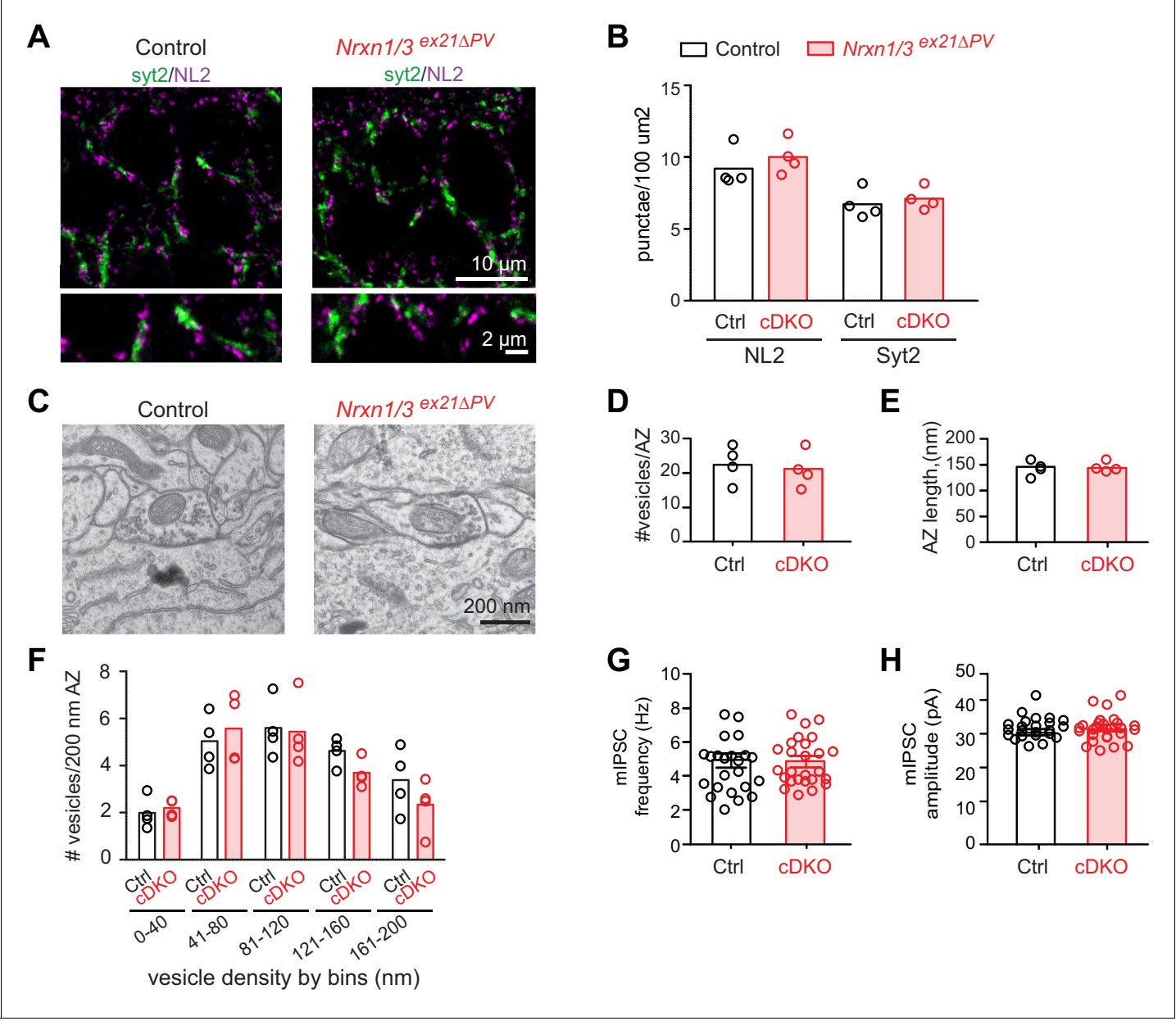

**Figure 5.** Analysis of PV+ specific *Nrxn1/3^{ex21ΔPV}* conditional knock-out mice. (**A**) The density of perisomatic GABAergic synapses in *stratum pyramidale* in CA1 of control and *Nrxn1/3^{ex21ΔPV}* PV conditional double knock-out (cDKO) mice was examined by immunohistochemistry with anti-Neuroligin2 (NL2, in magenta) and anti-Synaptotagmin2 (Syt2, in green) antibodies at postnatal day 24–26. Note that Syt2 immuno-reactivity is specific for presynaptic terminals of PV+ cell synapses whereas NL2 is a common postsynaptic marker for most GABAergic synapses. Thus, only a fraction of NL2 puncta is apposed to Syt2-positive terminals. (**B**) The density of NL2 and Syt2 puncta per 100 μm² of cell body area was quantified. Single dots in the graph represent means of respective synapse markers per animal (n = 4 mice per genotype). (**C**) Ultrastructure of inhibitory synapses in CA1 *stratum pyramidale* of littermate controls and *Nrxn1/3^{ex21ΔPV}* mice. Presumptive PV+ cell perisomatic termini were identified by the presence of large mitochondria in the synapses apposed on the membrane of the soma and by their morphology (*Takács et al., 2015*). (**D**) Average vesicle numbers per active zone and (**E**) average active zone length in nm. Single dots in the graph represent means per animal (n = 4 mice for each genotype, ≥76 synapses). (**F**) Average number of vesicles located in 40 nm bins with increasing distance from the active zone normalized to 200 nm active zone length. Single dots in the graph represent means per animal (n = 4 mice for each genotype, ≥76 synapses). (**G**) mIPSC frequency and (**H**) amplitude in control and *Nrxn1/3^{ex21ΔPV}* mice. The recordings were performed in parallel with littermate controls. Single dots in the graph represent single cells (n = 3 control and 4 *Nrxn1/3^{ex21ΔPV}* mice, mean ± SEM).

The following figure supplements are available for figure 5:

**Figure supplement 1.** Analysis of PV+ specific *Nrxn3^{ex21ΔPV}* single conditional knock-out mice.

*Figure 5 continued on next page*

*Figure 5 continued*

**Figure supplement 2.** Quantification of PV$^+$ synaptic termini on PV somata in the hippocampus.

from the majority of hippocampal CA1 interneurons marked in the *Pvalb^cre^::*Ai9$^{Tom}$ mouse line. Correspondingly, these cells generate high levels of AS4+ neurexin isoforms. It remains to be tested whether the absence of STAR-family RNA binding proteins is sufficient to direct alternative splicing to the inclusion of AS4+ or whether other, yet unidentified alternative splicing factors contribute to this alternative splicing choice. Finally, we demonstrate that this selective expression of AS4+ variants coincides with PV$^+$ cell enrichment of Cbln4, an AS4+ specific neurexin ligand. Interestingly, the differential expression of AS4 isoforms in PV *versus* principal cells is also in part observed for cortical cell populations (*Figure 2—figure supplement 1E*). In additional RiboTrap purifications from somatostatin-positive interneurons in the hippocampus we also observed an enrichment of *Nrxn AS4* + isoforms as compared to principal cells (*Figure 2—figure supplement 1F*) along with Slm2-negative somatostatin-expressing cells in the hippocampus (in particular in the hilus, *Figure 2—figure supplement 1D*). Thus, our study uncovers a hippocampal interneuron-specific gene expression program that consists of neurexin isoforms and ligands in the hippocampus.

## Functional consequences of cell type-specific splicing regulation

The difficulty of assigning cell type-specific *Nrxn* transcripts and splice isoform expression has been a major impediment for the interpretation of functional studies. Global knock-out studies demonstrated that neurexin alpha triple knock-out mice show a 40% reduction in the density of presumptive inhibitory synapses and defects in synaptic transmission in the brain stem of perinatal mice (*Missler et al., 2003*). Moreover, simultaneous global ablation of all *Nrxn3* transcripts results in brain area-specifc alterations in synaptic transmission (*Aoto et al., 2015*). In these studies it was unknown which endogenous primary transcripts and which splice isoforms are expressed and the knock-out manipulations were not cell type-specific. The data presented here demonstrate that pyramidal and PV$^+$ cells share primary *Nrxn1,2,3* transcripts but that the endogenous neurexin isoforms differ dramatically in their alternative splicing regulation at AS4.

In cellular assays, we demonstrate that the AS4+ splice insertion in Nrx3$\alpha$ selectively increases Nrx3 activity towards assembly of GABAergic postsynaptic structures and that AS4+ variants bind to the extracellular ligand Cbln4 that is co-expressed in PV$^+$ cells. In the mouse cerebellum, Neurexin-Cbln1 interactions have been demonstrated to be essential for the formation and stability of granule cell-Purkinje cell synapses (*Uemura et al., 2010*; *Ito-Ishida et al., 2012*). Similar to granule cells, PV$^+$ cells in the hippocampus express a Cbln family protein (Cbln4) and neurexin AS4+ isoforms which interact with Cblns. However, genetic disruption of *Nrxn1 and 3* AS4 insertions did not result in morphological alterations of PV$^+$ cell output synapses detectable at the level of light or electron microscopy. Structural functions of a Nrx1/3-Cbln4 pair, analogous to the cerebellar system, may be compensated by the remaining Nrx2 AS4+ isoforms or conditional ablation of the AS4 insertions with the *Pvalb^cre^* line may not be complete when PV$^+$ cell synapses form during postnatal development. Alternatively, the essential roles for Nrx1/3-Cbln4 complexes in PV$^+$ cells may differ from what is observed for Nrx-Cbln1 complexes in the cerebellum. Regardless, the increased c-fos$^+$ expression in the hippocampus of *Nrxn1/3^{ex21ΔPV}* mice indicates that the AS4+ isoforms have important functions and presumably reflects an impairment in PV$^+$ cell mediated inhibition in hippocampal circuits. Moreover, the impairment of short-term memory further supports the notion that neurexin AS4+ isoforms have important functions in the cells targeted by the *Pvalb^cre^* line in the hippocampus. Dysfunction of PV$^+$ interneuron networks has been postulated to be a major contributor to the pathophysiology of schizophrenia and autism (*Del Pino et al., 2013*; *Gogolla et al., 2014*). Thus, we speculate that part of the neuronal circuit deficits resulting from *Nrxn1 and 3* mutations in disease states may directly impair output synapses of the PV-interneuron network.

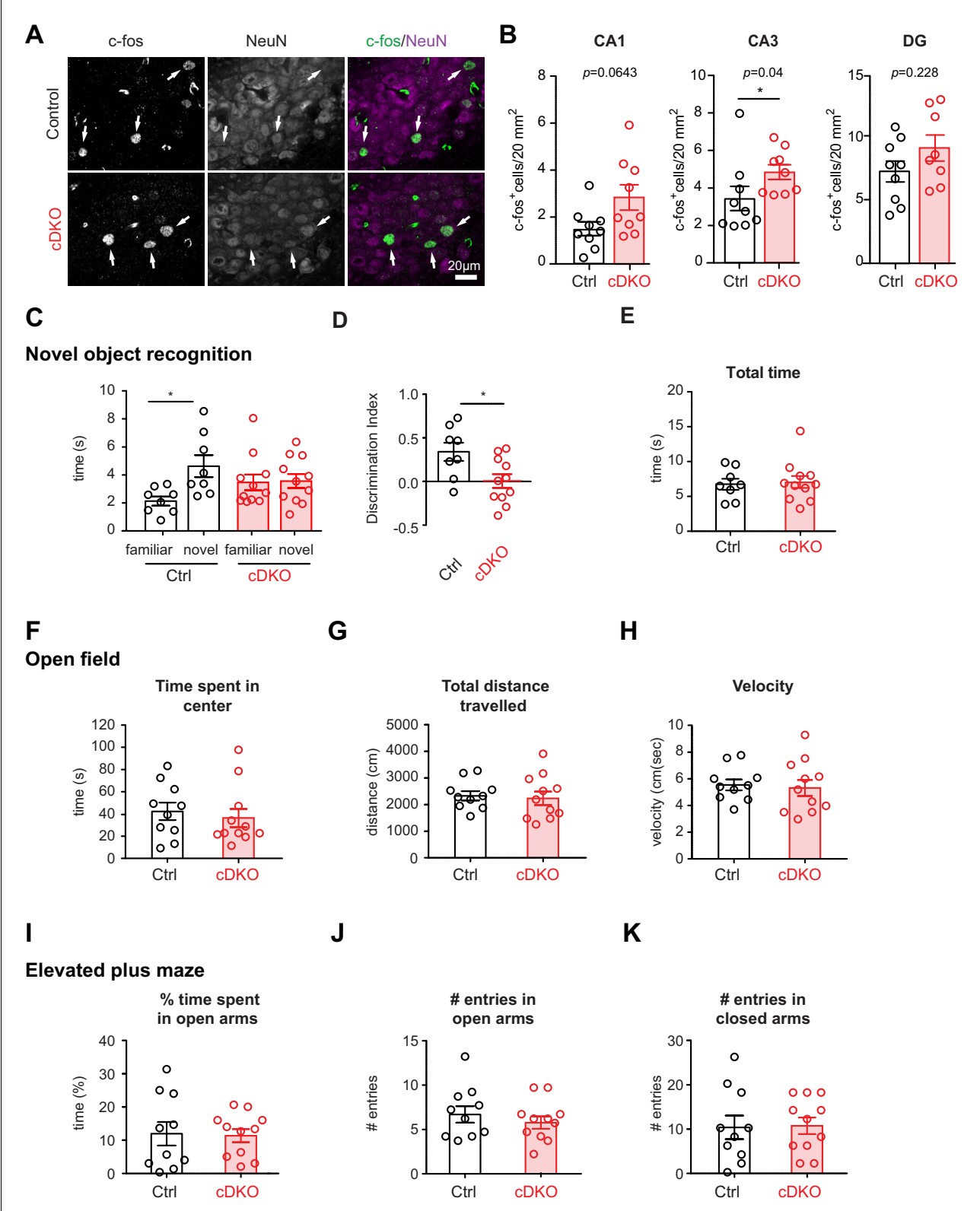

**Figure 6.** Analysis of PV hippocampal network activity in *Nrxn1/3^ex21ΔPV^* conditional knock-out mice. (**A**) Density of c-fos⁺ cells per 20 mm² in control and *Nrxn1/3^ex21ΔPV^* mice in the CA3 region. c-fos⁺ cells (green in overlay) co-localize with neuron-specific marker NeuN (magenta in overlay). Arrows indicate examples for c-fos⁺ neurons. (**B**) Quantification of control and *Nrxn1/3^ex21ΔPV^* c-fos⁺ neurons in CA1, CA3 and DG regions. Single dots in the graph represent mean per animal (n = 9 for control and *Nrxn1/3^ex21ΔPV^* mice, mean ± SEM, Mann Whitney test, ***p<0.001, **p<0.01, *p<0.05). (**C**)
*Figure 6 continued on next page*

*Figure 6 continued*
Novel object recognition test. Exploration time for familiar versus novel objects. Single dots in the graph represent single animals (n = 8 control and 11 *Nrxn1/3 $^{ex21\Delta PV}$* mice, mean ± SEM, Mann Whitney test, p=0.0086, \*\*\*p<0.001, \*\*p<0.01, \*p<0.05). (D) Discrimination index for familiar versus novel object for the same animals as in (C) (mean ± SEM, Mann Whitney test, p=0.285, \*\*\*p<0.001, \*\*p<0.01, \*p<0.05). (E) *Nrxn1/3$^{ex21\Delta PV}$* mice do not differ in the total object exploration time in the object recognition assay. Single dots in the graph represent single animals (mean ± SEM). (F–H) Locomotor activity of *Nrxn1/3$^{ex21\Delta PV}$* mice in open field. Mice were allowed to explore a square arena (50 cm x 50 cm x 25 cm) for 7 min. (F) The time in center, (G) the total distance traveled and (H) velocity are quantified. Single dots in the graph represent single animals (n = 10 control and 11 *Nrxn1/3 $^{ex21\Delta PV}$* mice, mean ± SEM). (I–K) In the elevated plus maze *Nrxn1/3 $^{ex21\Delta PV}$* show similar behavior as cre-negative littermate control mice. Single dots in the graph represent single animals (n = 10 control and 11 *Nrxn1/3 $^{ex21\Delta PV}$* mice, mean ± SEM).

## Materials and methods

### Mice

All procedures involving animals were approved by and performed in accordance with the guidelines of the Kantonales Veterinäramt Basel-Stadt (Licences 2272, 2293, 2599). *Nrxn3$^{ex21\Delta}$* conditional knock-out mice were generated in collaboration with Dr. Siu-Pok Yeh at the University of Connecticut Gene Targeting Facility. The mice carry loxP sites on either side of exon 21 of the *Nrxn3* gene. A targeting vector was designed with a loxP site placed 718 bp upstream of exon 21 of mouse *Nrxn3* and a loxP-Frt-PGKneo-Frt cassette inserted 840 bp downstream of the same exon. The 5' and 3' homology arms were approximately 5 kb and 3 kb in length, respectively, and the targeting construct contained herpes simplex virus thymidine kinase as negative selectable marker in the 3' arm of the targeting vector. The construct was electroporated into ES cells derived from F1(129Sv/B6) embryos. Correctly targeted ES cells were then used to generate chimeric mice. Male chimera were mated with ROSA26-Flpe females (Jax stock no: 003946), which had been backcrossed over five generations with C57Bl6J, to remove the PGKneo cassette. The resulting heterozygotes were intercrossed to produce homozygotes. Mice that are homozygous for this allele are viable, fertile, normal in size and do not display any gross physical or behavioral abnormalities. Germline ablation was created by crossing with CMV-cre mice (Jax stock no: 006054)(*Schwenk et al., 1995*). Nrxn1$^{ex21\Delta}$ conditional knock-out mice were previously described (*Traunmüller et al., 2016*). *Rpl22-HA* (RiboTag) mice (*Sanz et al., 2009*), *Pvalb$^{cre}$* mice (*Hippenmeyer et al., 2005*), *SST$^{cre}$* mice (*Taniguchi et al., 2011*), *CamK2$^{cre}$* mice (*Tsien et al., 1996*) and Ai9$^{Tom}$ (*Madisen et al., 2010*) mice were obtained from Jackson Laboratories (Jax stock no: 011029, 017320, 005359, 007909 respectively).

### In situ hybridizations

In situ hybridizations were performed as previously described (*Schaeren-Wiemers and Gerfin-Moser, 1993*). DNA constructs encoding probes for primary neurexin transcripts contained SP6 and T7 promoters at 5'- or 3'-end, respectively:

*Nrxn1α*:TATCG*ATTTAGGTGACACTATAGA*AGAAGGGGAATAACCAGGCAGATTTCTGAGTTC
TTATAAACGTCAAAATAAAATCAAAATGCATCACCAACATACAACCAACTGACATGCTGGAGTC
TTAGGGTGGCCTTGCAGAGACAGAAAAAAGAAGAGTTTGAGGGATTGATTTCACCATCTAACAGC
TTCTGGCTACTTCACAGCAGGGTTTGAGACCTACCTACAGGGCTCCTAAACATAAAGCGAA
TCAGCCTGGGAGGGCATGCACAGGAAATTTGGCCTTGGCTTTAGTGGTGCTGGAAGCCCATGA
TATGACAGAAGTAGGCCTCTGAAGCTATTCTGGTGTCTCACTCTGTCCTTTCTCCTTCTGTGC
TTGGAGCCAACATCTGGAGGGGCCTCGAGGCCTGCCAAGCAGCCAAAGATCCTGGCTCAGTG
TTGACTTCGTGTTCCTCTCCGAGAAAAGTGTGTTTCTTCCCTATAGTGAGTCGTATTACAATTC

*Nrxn1*:CTATCG*ATTTAGGTGACACTATAGA*AGGCTCTTTCATCTGCCCCTGCTTTTCCTCCGC
TCGCTTTCCCCAGTTCGATCCCTGCTGTCTTCACGAGGGTGCCCACTTCCCTCTGAACCCATCG
TCGGGCGTAGTGTCAGGAGGCGGCGGACTCGGAGATTGCCTCTGGAGCAGGCGA
TGCGCGCCGCTGCTCTGCGCGCTGCCCGGGTGAGGCTGGCGGGAGCTGGAGAGC
TGGCCAGGGCTGAATGGAGGGACAGGGTGCCTTGCGTCCATGGGTCTGCTTCTTTCCTGAAAG-
GAGGCTGGACCGGCGAAGTGGTCTCCCAGTTCCCCGCGCACAATGCTAAATGGATTTACCTAG
TGGATTCCCGGTGGATGGCTGTCATGTAGAAGTGAAGACCCTCCGGGAGGAGCTTTAAACAA
TTTCCAGGCTCCCCAACCCCGGCACACACTCTCGCCCGAAACTCTTGGGGAGTATCCCTATAG
TGAGTCGTATTACAATTC

*Nrxn2α*:CTATCG*ATTTAGGTGACACTATAGA*AGACCAGGAGGCGCGAGGCAGCCGATATCGC TGGCCCAGGATCTGTTACCTGCCGTAGACCCAGCGGTCTCTAGGCTCGGATCCCTACCCTTCAGC TCCTGGCGCCCCCAAAACCAGGCGTCCCTCCCCCACCTTCCATACGAGCCCGCCGCGGGG- GAGGGGCTCCACCACCGCAGCCGCCGTCGTTGCCTTCCGGGGACGTGGACACG TGAGCCCCGGCTACTGAGTCCATGGCACTGTGAATCGGCGAGGTCCCTATAGTGAGTCGTATTA- CAATTC

*Nrxn2*:CTATCG*ATTTAGGTGACACTATAGA*AGTGAGGGGGGGACCCCTAGCCGCCCGCGA TGGATCCAGGCTTCACGGACCTTGGCCTTCCCGCTGCGCGTACCCCGGATTCCCCGGCGGGA TCCAGTTGATTTGCTTGGCTCCGGACTGAGGCTCGGGCTCTGGTTTTCCTTCGCTTCACCCC TACCCCCCTCTCGGAGCTCGCAACCGGAGGGGGGCTTTCCCTATAGTGAGTCGTATTACAATTC

*Nrxn3α*:CTATCG*ATTTAGGTGACACTATAGA*AGGAAGTCTGAATCTGCTTCCGCTCTGCTC TGGGCCTCACTCCACCTGAGTCCTCAGTTGTTTGCGGTTCCCCTTCCCAGGGTCTGCTGC TAGACCGTCAAACCTCAGCACTGGGCCTTGGCTTGGGCCTGCCTTTTGCCTGGCTCACCTCCCGA TTACTCCTCCTCCATTACAGCACCCTGAGCCCCAGCCCTGTCCTTGGTCTTCCTGGCTAGGACGCA TTTGCCGGGAGGAAGACATTACGGAAGGCTTATTCCCACCCTGGGCTCCTTCTCCTCCTTGAA TCAAGGCCTCCGGATCCACATGGATAGCTGAGATCTTTTCTTGGAGAAAGATACTTCTTCCTCGCC TCATCCCTGATTTGCCTCACCCGACAAATCCCCTGTCTGTTTCGTCTCCCTCTTTATGGGATTTC TTGCTTGTGTGCCTATCTAGGGCCGTGTTGTCCTCCCTATAGTGAGTCGTATTACAATTC

*Nrxn3*:CTATCG*ATTTAGGTGACACTATAGA*AGCTGCTCCTCTCCACCTTCTGCTACGTTGGTC TGGGTGGCTAGCTCAGTGCTGTTTCTTTTCCTCTGGCCCTTCTTGATCCCTTTCTCTGGCTACTGC TGCTGGCTGATTTTCAACCTATTGGGAACTCAGGACTTAAGGCGGCTGCACCGTGGCGCTCA TCCAGGACACTCAGGGTTAACAGCCTCCGCGCCCATCCACAGAGACTCCCGGGAGCAGAACTC TTCCACCTGCAGCCCCCTTTTGCCTGGCAGTTCTGCATTGCATCGCTTGGAAGTCGAAACAA- GAAAGAAAGAAAGAAAGAAAAATGTCCAAACTCCTTGGATGTTGGGAAAAACTAGCGTGAA TTTACTTGGTTTTTTCCTGCTCTGTCTTCTCTCTCCGTTTCCACCTTTCCCCAGTGGCTTTCCAGAG TATGCAGCTAGTACATCGGACTTGAAGTACCAAGTCCCTATAGTGAGTCGTATTACAATTC

Templates for in vitro transcription using SP6-polymerase (sense probe) or T7-polymerase (anti-sense probe) were amplified by PCR using ISP-SP6-5' (5'-CTATCGATTTAGGTGACACTATAGAAG-3') and ISP-T7-3' (5'-GAATTGTAATACGACTCACTATAGGGA-3') primers.

For the dual labeling of *Nrxn3α* in PV[+] cells in CA1 of *Pvalb[cre]*::Ai9[Tom] mice, the in situ *hybridization* with *Nrxn3α* probes was performed first. To identify the td Tomato PV[+] cells, the brain sections were incubated with anti-RFP. The imaging was done using a Nikon Eclipse E800 (Plan Apo 40X/ 0.95)

## Antibodies and SRM assays

Custom antibodies to Slm2 were previously described (*Iijima et al., 2011*). The following commercially available antibodies were used: anti-c-Fos (K-25, Santa Cruz), anti-Gapdh (D16H11, Cell Signaling), anti-Gephyrin (mAb7a, Synaptic Systems), anti-GFP (affinity purified, homemade), anti-HA (11867431001, Roche), anti-His6 (A190-114A, Bethyl), anti-Myc (9E10, Invitrogen), anti-NeuN (MAB377, Chemicon), anti-Neuroligin2 (sc-1487, Santa Cruz), anti-pan Nrx (affinity purified, homemade), anti-Parvalbumin (PVG214, Swant), anti-PSD95 (51–6900, Invitrogen Novus), anti-RFP (600-401-379, Rockland) anti- Synaptotagmin2 (Znp1, Zebrafish International Resource), anti-Tubulin (E7, Developmental Studies Hybridoma Bank),anti-V5 (C-9, Santa Cruz).

Selected reaction monitoring (SRM) for quantification of neurexin protein levels was performed as previously described (*Schreiner et al., 2015*).

## Molecular biology assays

For Ribotag purifications the procedure of Heiman and colleagues for affinity-purification of polysomes (*Heiman et al., 2014*) was modified as follows:

Brains of mice between (postnatal day 24 to 28) were dissected in ice-cold PBS and four hippocampi (two animals per condition) were lysed in 500 µL of homogenization buffer containing 100 mM KCl, 50 mM Tris-HCl pH 7.4, 12 mM MgCl$_2$, 100 ug/mL cycloheximide (Sigma), 1 mg/mL heparin (Sigma),1x complete mini, EDTA-*free* protease inhibitor cocktail (Roche), 200 units/mL RNasin© plus inhibitor (Promega) and 1 mM DTT (Sigma). The lysate was centrifuged at 2'000xg for 10 min. Igepal-CA380 was then added to the supernatant to a final concentration of 1%. After 5 min incubation

on ice, the lysate was centrifuged at 12'000xg for 10 min. Anti-HA coupled magnetic beads (Pierce) were added to the supernatant in the following concentrations: 25 µL/mL for CamK2$^{Ribo}$ and 15 µL/mL of beads for PV$^{Ribo}$ samples, respectively. Incubation was performed at 4°C for 4 hr. The beads were washed four times in washing buffer containing 300 mM KCl, 1% Igepal-CA380, 50 mM Tris-HCl, pH7,4, 12 mM MgCl$_2$, 100 µg/mL Cycloheximide (Sigma) and 1 mM DTT(Sigma). The beads were eluted in 350 µL of RLT plus buffer (Qiagen). The RNA purification was performed using RNeasy mini plus kit following the manufacturers' instructions. 30–50 ng of total RNA was reverse transcribed using random hexamers and ImProm II Reverse Transcriptase (Promega).

For the misexpression of Slm2, granules cells were infected with either AAV encoding for Slm2-2A-Venus YFP or GFP containing the human synapsin promoter at day in vitro three and were harvested at day in vitro 14.

Quantitative PCR was performed on a StepOnePlus qPCR system (Applied Biosystems). To assess *Nrxn* expression level gene expression assays (see *Table 1*, *Table 2*) were used with TaqMan Fast Universal Master Mix (Applied Biosystems) and comparative C$_T$ method. The mRNA levels were normalized to *β-actin* mRNA or to pan-*Nrxn* mRNA. To determine the enrichment fold of mRNA purification, DNA oligonucleotides were used with FastStart Universal SYBR Green Master (Rox) (Roche) and comparative C$_T$ method (see *Table 3*). For each assay, two technical replicates were performed and the mean was calculated.

Standard PCR reactions were performed using 5X Firepol Master mix (Solis BioDyne). Radioactive semi-quantitative PCR was performed with γ-$^{32}$P 5' end-labelled primers (Hartmann) using 5X Firepol mastermix (Solis Biodyne). For the quantification of the radioactive semi-quantitative PCR (see *Table 4*), the gel image was acquired with Typhon FLA 700 (GE Healthcare). The PCR band intensity was analyzed using Fiji software. The background intensity was subtracted from the mean intensity of each PCR band and then divided by the number of GTP and CTP in PCR product (Mean$_{Norm}$). To calculate the % of inclusion the following formula was used: Mean$_{NormAS+}$/[(Mean $_{NormAS4+}$)+(Mean $_{NormAS4−}$)].

## Immunohistochemistry

Animals (male and female) from postnatal day 24 to 26 were transcardially perfused with fixative (4% paraformaldehyde/15% picric acid in 100 mM phosphate buffer, pH 7.4). The brains were post-fixed overnight in same fixative at 4°C and incubated in 30% sucrose in PBS for two nights at 4°C and frozen at −80°C. Tissue was sectioned at 30 µm on a cryostat for synapses quantification and 35 µm for HA. Floating sections were immunostained following standard procedures for HA immunostaining. For synapse quantitative analysis, sections were treated with pepsin before staining, as described (*Panzanelli et al., 2011*). Briefly, sections were incubated in prewarmed 0.1M phosphate buffer at 37°C for 10 min and then in 0.15 mg/mL pepsin (Dako) dissolved in 0.2M HCl for 10 min. Images for synapse quantification were acquired at room temperature on an upright LSM700 confocal microscope (Zeiss) using 63x Apochromat objectives and controlled by Zen 2010 software.

For quantification of synaptic markers, three to four brain sections were used per genotype. Additionally, four to six confocal planes within the *stratum pyramidale* of CA1 were acquired per section. Using Metamorph software (Molecular Devices) area of somata were manually drawn to measure the area. Synaptotagmin2 and perisomatic Neuroligin2 punctae were counted manually in separated channels and normalized to 100 µm2. For each genotype the average synapses punctae for each animal was calculated. Control littermates mice carried the floxed alleles but were Cre negative (n = 6 mice for control and five mice for *Nrxn3$^{ex21ΔPV}$* and four mice each for control and *Nrxn1/3$^{ex21ΔPV}$*).

**Table 1.** Commercially available gene expression assays for *Gapdh, Nrxn1,2,3* (Applied Biosystems).

| *Gapdh* | **Mm99999915g1** |
| --- | --- |
| *Nrxn1 pan* | Mm00660298_m1 |
| *Nrxn2 pan* | Mm01236851_m1 |
| *Nrxn3 pan* | Mm00553213_m1 |

**Table 2.** Custom gene expression assays from TIB Molbio (Berlin).

| *Nrxn one alpha*-F | **5'- CAG CAC AAC CTG CCA AGA −3'** |
| --- | --- |
| *Nrxn one alpha*-R | 5'- GTC CCA GGG TCA TTG CAG A −3' |
| Probe *Nrxn1 alpha* | 6FAM-TGG GCC ACT GAA GGA AGT CAT GCT–BBQ |
| *Nrxn one beta*-F | 5'- CCT GGC CCT GAT CTG GAT AGT −3' |
| *Nrxn one beta*-R | 5'- TTG TCC CAG CGT GTC CG −3' |
| Probe *Nrxn1 beta* | 6FAM-CTG AAT GAT GCT TGC TGC TGC CA -BBQ |
| *Nrxn two alpha*-F | 5'- CAC CAC CTG CAC CGA AGA G −3' |
| *Nrxn two alpha*-R | 5'- CCG GAG GCA CTG TCC ACT-3' |
| Probe *Nrxn2 alpha* | 6FAM- CCC CCT TCC CGA AGA TGT ATG TGG TCC -BBQ |
| *Nrxn two beta*-F | 5'- GTG CCC ATC GCC ATC AA −3' |
| *Nrxn 2beta*-R | 5'- TTG GAG GCG TTC ATT ATC AGT GTT −3' |
| Probe *Nrxn2 beta* | 6FAM- CCC CCT TCC CGA AGA TGT ATG TGG TCC -BBQ |
| *Nrxn three alpha*-F | 5'- CTG TGA CTGCTC CAT GAC ATC ATA TT −3' |
| *Nrxn three alpha*-R | 5'- CA GAG CGT GTG CTG GGT CT-3' |
| Probe *three alpha* | 6FAM- CGC TTT TCC CAA AGA TGT ATG TTG CAC CA -BBQ |
| *Nrxn three beta*-F | 5'- AAG CAC CAC TCT GTG CCT ATT TCT −3' |
| *Nrxn three beta*-R | 5'- CCA GGG GCG CTG TCA AT-3' |
| Probe *Nrxn3 beta* | 6FAM- CGC TTT TCC CAA AGA TGT ATG TTG CAC CA -BBQ |

The quantification of the number of PV⁺ cells was also done using an upright LSM700 confocal microscope (Zeiss) using 10X objectives. Stacks of 22 μm with (2.2 μm step size) were acquired.

Images for assessing the Rpl22-HA expression in the CamK2$^{Ribo}$ and PV$^{Ribo}$ were acquired at room temperature on an inverted LSM500 confocal microscope (Zeiss) using 10x and 20 x objectives.

To quantify the number of PV inputs on PV⁺ interneurons, brain sections from mice aged between 5 to 8 weeks were stained for Synaptotagmin2 and Parvalbumin (n = 4 mice for each genotype). The images of dorso-medial hippocampi were acquired on a LIS-spinning disk confocal system (40X/1.3 NA objective, 0.1 μm step size). Four separate fields of the hippocampal regions of interest were acquired in two separate brain sections per animal. The density of Synaptotagmin2 on PV⁺ interneurons was manually counted using Fiji software. The average synapses punctae for each animal was calculated.

For Slm2 immunolabelling, animals (male and female postnatal day 25–30) were transcardially perfused with fixative (4% PFA in 100 mM Phosphate Buffer, pH = 7.2). Tissue was sectioned at 50 μm in PBS on a vibratome. Images of dorso-medial hippocampi were acquired at room temperature on a confocal microscope using 40X Apochromat objectives, which were controlled by Zen 2010 software (1 μm step sizes). For assessment of Slm2 expression in *Pvalb$^{cre}$*::Ai9$^{Tom}$ (n = 5 mice) and *SST$^{cre}$*::Ai9$^{tom}$ (n = 4 mice) positive interneurons, Imaris was used. Briefly, Slm2 intensity levels were characterized in tdTomato-positive cells and clustered into Slm2 negative, high and intermediate levels. Assessment of Slm2 expression in pyramidal cells was done in the dorso-medial hippocampi of CamK2$^{Ribo}$ animals (n = 4 mice). Hippocampal slices were stained for HA (for Rpl22-HA expression) and Slm2. Overlap between Rpl22-HA and Slm2 was determined by analysis of single planes in Fijii (Image J). Slm2 background intensity levels were determined by Slm2 immunostaining in the dentate gyrus and *Slm2$^{KO}$* mice.

For analysis of overlap between *Pvalb$^{cre}$*::Ai9$^{Tom}$ positive interneurons and antibody labelled PV neurons, dorso-medial hippocampal slices were stained for Parvalbumin (n = 5 mice, postnatal day 25–30). Co-labelling of genetic td Tomato and the PV antibody was assessed in stacks of ~30 μm (1 μm step sizes) with Fijii (Image J).

Images were assembled using Adobe Photoshop and Illustrator Software.

**Table 3.** DNA Oligonucleotides used with SYBR Green-based real-time PCR.

| *CamK2*-F | 5'- GAG GAA CTG GGA AAG GGA G −3' |
| --- | --- |
| *CamK2*-R | 5'- GGT AAC CTA CCT CTG GCT G −3' |
| *Cbln2* -F | 5'- AGA CAA ACT ATC CAG GTC AGC −3' |
| *Cbln2*-R | 5'- CCT GGT AAC ATC CTG GTC TC −3' |
| *Cbln4*-F | 5- TTTGATCAGATCCTGGTTAACG −3' |
| *Cbln4*-R | 5- ACTATATTCCTTTCCTCGGT −3' |
| *Erbb4*-F | 5'- ATC CCT GTG GCT ATA AAG ATC C −3' |
| *Erbb4*-R | 5'- CAT GAT CAG AGC CTC ATC CA −3' |
| *Gfap*-F | 5'- CTC GTG TGG ATT TGG AGA G −3' |
| *Gfap*-R | 5'- AGT TCT CGA ACT TCC TCC T −3' |
| *Gad1*-F | 5'- GTA CTT CCC AGA AGT GAA GAC −3' |
| *Gad1*-R | 5'- GAA TAG TGA CTG TGT TCT GAG G −3' |
| *Pvrl3*-F | 5'- GTG ACT GTG TTA GTT GAA CCC −3' |
| *Pvrl3*-R | 5'- TGC TAC TGT CTC ATT CCC TC −3' |
| *Pvalb*-F | 5'- CATTGAGGAGGATGAGCTG −3' |
| *Pvalb* -R | 5'- AGTGGAGAATTCTTCAACCC −3' |
| *Vglut1*-F | 5'- ACC CTG TTA CGA AGT TTA ACA C −3' |
| *Vglut1*-R | 5'- CAG GTA GAA GGT CCA GCT G −3' |
| *Sam68*-F | 5'- GGG AAG GGT TCA ATG AGA GA −3' |
| *Sam68*-R | 5'- AAT GGG CAT ATT TGG GGT CT −3' |
| *Slm1*-F | 5'- GAC CAA GAG GAA ACT CCT TGA A −3' |
| *Slm 1* R | 5'- GGC ATG ACT CAT CCG TGA ATA −3' |
| *Slm2*-F | 5'- GGT CCG CGT GGC AAT TC-3' |
| *Slm2*-R | 5'- CAT CCG GGC ATA TGC TTC T-3' |
| *Wsf1*-F | 5'- CATCATTCCCACCAACCTG −3' |
| *Wsf1*-R | 5'- TAC TTC ACC ACC TTC TGG C −3' |

Statistical analyses were done with Prism software (Graphpad software). The image acquisition and analysis were done blinded with respect to the genotype of the animals.

## Electron microscopy

Animals (postnatal day 24–25, *Nrxn3$^{ex21\Delta PV}$*, *Nrxn1/3$^{ex21\Delta PV}$* and respective control littermates which carried the floxed alleles but were Cre negative) were transcardially perfused with fixative (2% para-formaldehyde, 2% glutaraldehyde in 100 mM phosphate buffer [pH 7.4] and brains were postfixed for 1 hr. Tissues were sectioned coronally at 60 µm thickness in PBS on a vibratome. Sections from the dorso-medial hippocampi were analyzed for each genotype. Sections were washed in 0.1 M cacodylate buffer [pH 7.4], postfixed in 0.1 M reduced osmium (1.5% $K_4Fe(CN)_6$, 1% $OsO_4$ in water) and embedded in Epon resin. Images were acquired on a Transmission Electron Microscope (Fei Morgagni, 268D). Quantification of the number and distribution of vesicles was performed using Reconstruct software (http://synapses.clm.utexas.edu/tools/reconstruct/reconstruct.stm). All image acquisition and analysis was done blinded with respect to the genotype of the animals. Independent data sets were collected from four mice for each genotype. PV-terminals were identified by their localization on the neuron plasma membrane and the presence of large mitochondria. The vesicles and the active zones were manually drawn. The shortest distance from the vesicle membrane to the active zone membrane was then calculated and all vesicles at distances of less than 200 nm were taken into account.

**Table 4.** DNA Oligonucleotides used for radioactive semi-quantitative and standard PCR.

| *Nrxn1 AS2*-F | 5'- tgg gat cag ggg cct ttg aag ca-3' |
|---|---|
| *Nrxn1 AS2*-R | 5'- gaa ggt cgg ctg tgc tgg gg −3' |
| *Nrxn2 AS2*-F | 5'- gca cga cgt ccg ggt tac cc-3' |
| *Nrxn2 AS2*-R | 5'- ggt cgg ctg tgt tgg ggc tg −3 |
| *Nrxn3 AS2*-F | 5'- tcc ggg gcc ttt gag gcc at-3' |
| *Nrxn3 AS2*-R | 5'- gcg gta ctt ggg ctt cca cca-3' |
| *Nrxn1 AS3*-F | 5'- tgg agc tag atg cag gac gtg tga a −3' |
| *Nrxn1 AS3*-R | 5'- ttc ctc gcc gaa cca cac gc −3' |
| *Nrxn2 AS3*-F | 5' – ccc act cgc atg cac acg ga −3' |
| *Nrxn2 AS3*-R | 5'- tgc ccc gca aac agt gtc tcg −3' |
| *Nrxn3 AS3*-F | 5'- tgt cac agc gag cct atg ggc −3' |
| *Nrxn3 AS3*-R | 5'- tct ccg cac tac ccg gac gg −3' |
| *Nrxn1 AS4*-F | 5'- ctg gcc agt tat cga acg ct −3' |
| *Nrxn1 AS4*-R | 5'- gcg atg ttg gca tcg ttc tc −3' |
| *Nrxn2 AS4*-F | 5'- caa cga gag gta ccc ggc −3' |
| *Nrxn2 AS4*-R | 5'- tac tag ccg tag gtg gcc tt −3 |
| *Nrxn3 AS4*-F | 5'- cca gga atg ggg gaa atg ct −3' |
| *Nrxn3 AS4*-R | 5'- ttg tcc ttt cct ccg atg gc −3' |
| *Nrxn1 AS6*-F | 5'- ctg gcc agt tat cga acg ct −3' |
| *Nrxn1 AS6*-R | 5'- gcg atg ttg gca tcg ttc tc −3' |
| *Nrxn1 AS6*-F | 5'- ctg gcc agt tat cga acg ct −3' |
| *Nrxn1 AS6*-R | 5'- gcg atg ttg gca tcg ttc tc −3' |

## Hippocampal neurons / fibroblast co-culture assays

Co-culture assays were performed essentially as previously described (*Scheiffele et al., 2000*). Hippocampal neurons isolated from E16 mouse embryos were seeded at a density of 40.000 cells / well (24 well plates) on cover slips coated with poly-D-lysine. At DIV19 HEK293 (ATCC, Cell line was used within 2 years of being purchased from ATCC. No Mycoplasma contamination was detected in bi-annual tests of the cultures) cells transfected with neurexin-alpha constructs (with or without insertion at AS4) were added at a density of 6.000 cells / well. Three days after co-culture mixed cultures were fixed with 4% PFA/4% Sucrose in PBS for 10 min at RT, washed 2x with PBS, and blocked and permeabilized in blocking solution (10% normal donkey serum/2% BSA/0.1% Triton-X100 in PBS) for 1 hr at RT. Blocking solution was removed and cover slips were incubated with primary antibodies anti-Gephyrin (mAb7a, Synaptic Systems) diluted 1:2000 in blocking solution overnight. Subsequently cover slips were washed twice with PBS and incubated with secondary antibodies (anti-mouse Alexa-564/1:750 in blocking solution) for 2 hr at room temperature, washed 4x with PBS and mounted on glass slides for microscopy.

Pictures (40x per experiment and condition) were acquired with a Nikon Eclipse E800 with an 60x objective (Plan Apo 60XA/1.40, oil) from fields containing GFP positive cells (transfected HEK-cells) and analyzed with ImageJ software as followed: (i) gephyrin positive areas co-localized with GFP-positive area were extracted using Colocalization Highlighter plugin, (ii) extracted gephyrin positive area as well as GFP positive area were calculated using particle analysis plugin, (iii) percentage of gephyrin positive area to total GFP area were calculated.

## Cbln binding and surface retention assays

For Cbln binding assay COS7 (ATCC, Cell line was used within 2 years of being purchased from ATCC. No Mycoplasma contamination was detected in bi-annual tests of the cultures) cells grown in six well plates in DMEM medium supplemented with 10% FBS were transfected with full-length HA-

Nrx1α, HA-Nrx3α (with or without insertion at the AS4), HA-Nrx or mock transfected. 24 hr post-transfection cells were trypsinized and distributed on 48-well plate (six well each construct / plate). On the next day cells were washed with warm DMEM. Conditioned medium containing Cbln1, 2 and 4 was collected from HEK293 cells and were added to Nrx-expressing cells. Plates were incubated for 3 hr at 37°C, washed 3x with cold DMEM and fixed with 4%PFA for 10 min. Fixed cells were washed 1x with PBS and blocked with 5% milk in PBS for 1 hr. After blocking cells were washed with PBS and incubated with primary antibody (mouse anti-V5-probe, or rat anti-HA, 1:2000, Roche) diluted in 1% dry milk in PBS at 4°C overnight. On the next day cells were washed 3x with PBS and HRP-conjugated antibody (1:2000, anti-Mouse-HRP or 1:4000, anti-Rat-HRP for detection of V5 or HA-antibodies, respectively) was added for 2 hr at RT. Subsequently, cells were washed 4x with PBS, and 100 µl/well Ultra-TMB substrate (Pierce) was added to the well. The reaction was stopped by addition of 1N NaCl, and the absorption was measured at 450 nm in an iMark microplate reader (Biorad).

For Cbln surface retention assays COS7 cells were co-transfected with HA-tagged Nrx1/3-full – length and Cbln-Myc constructs. 24 hr post-transfection cells were trypsinized and distributed on 96 well plates (six well each construct / plate). On the next day cells were fixed and cell surface bound Cbln and surface expressed Nrx were probed as described.

## Electrophysiology

For acute slice recording, postnatal day 23–30 mice were anesthetized with isoflurane and rapidly decapitated. Three hundred micrometer thick sagittal sections were cut in sucrose substituted artificial cerebrospinal fluid (ACSF) that consisted of 87 mM NaCl, 2.5 mM KCl, 1.25 mM NaH2PO4, 25 mM NaHCO3, 25 mM glucose, 75 mM sucrose, 0.5 mM CaCl2, 7 mM MgCl2. Slices were allowed to recover at 34°C for 1 hr and then maintained at room temperature in the same sucrose ACSF. For whole-cell recordings, slices were perfused with 125 mM NaCl, 2.5 mM KCl, 1.25 mM NaH2PO4, 25 mM NaHCO3, 2 mM CaCl2, 1 mM MgCl2, 25 mM glucose, 4 uM AP5, 2 uM GYKI. For all experiments, whole-cell recordings were digitized at 10 kHz and filtered at 2 kHz. Whole-cell patch-clamp recordings of CA1 pyramidal cells were done using 2.7–3.5 U pipettes and filled with an internal solution that contained 170 mM CsCl, 10 mM Hepes, 0.5 mM QX314, 2 mM Mg-ATP, 0.5 mM NaGTP, 2 mM EGTA, 318 mOsm, pH 7.27. The cells were held at a holding potential of –70 mV. For mini recordings, slices were also perfused with 500 nM TTX. The mIPSCs were detected using Igor and the mIPSCs were detected using macros written by Dr. Taschenberger and modified by Dr. Kochubey.

## Behavioral assays

Behavioral testing was done with littermate control (carrying floxed alleles but were Cre negative) and $Nrxn1/3$ $^{ex21\Delta PV}$ males which were aged between 6 and 15 weeks. Behavioral tests were done with two mouse cohorts (five mice per genotype in each cohort). Before each behavioral test, mice were allowed to acclimate in the behavioral room for at least 30 min. In each tasks, arena and objects were cleaned with 70% ethanol between trials.

One day before the novel object recognition test, mice were placed in an open field arena, where they were allowed to explore freely 7 min the arena. Explorative behaviors were recorded by a Noldus Camera. Using the video tracking software Ethovision 10, the distance traveled, velocity and time spent in the center of arena were determined. This experiment was used as a habituation for the novel object recognition test.

For the novel object recognition test, mice were placed in a squared arena (50x50×25 cm). The recordings were acquired with a Canon camera. Animals were allowed to explore two identical small Falcon tissue culture flasks filled with sand for 5 min. After a 1 hr inter-trial interval, one flask was replaced with a tower of Lego bricks and duration of interaction was assessed in a 5 min trial. Mouse behaviors were acquired with a Canon camera and exploration time for each object was measured manually. Preferences for the novel object was expressed as a Discrimination Index (DI): (time $_{novel\ object}$− time $_{familiar\ object}$)/(time $_{novel\ object}$+time $_{familiar\ object}$). If a mouse exhibited less than 2 s in the total time exploring the objects (time $_{novel\ object}$+time $_{familiar\ object}$), it was excluded from the analysis. Object exploration was defined as the orientation of the mouse snout toward the object, sniffing or touching with the snout within 2 cm proximity. Leaning, climbing, looking over or biting the objects

were not considered as exploration time. The position of the objects in the test was counterbalanced between the animals in a group.

For the elevated plus maze, mice were placed at the junction of four arms (35 cm x 6 cm, 74 cm above the ground) of the maze and the behavior was recorded by a camera (Canon) for 5 min. Entries and duration in each arm were measured manually.

## Acknowledgements

We thank members of the Scheiffele Lab for support and constructive discussions. We are grateful to Siu-Pok Yee and the University of Connecticut Gene Targeting Facility for help with the generation of conditional knock-out mice. We thank Andrea Gomez for initial electrophysiology recordings, Elisabetta Furlanis for sharing RNA preparations and Oriane Mauger for data analysis advices. We also thank Markus Rüegg for kindly sharing *CamK2^cre* animals and Ursula Sauder for excellent assistance with the ultrastructural analysis. TMN was supported by a fellowship from the Werner-Siemens Foundation/International PhD program Fellowship for Excellence. DS was financially supported by a grant from the Forschungsfond of the University of Basel and a FP7 Marie-Curie Mobility Fellowship from the FP7 of the European Union. LT was supported by the Boehringer Ingelheim Fonds. This work was supported by funds to PS from the Swiss National Science Foundation, the National Competence Centre for Research NCCR-SYNAPSY, the European Research Council (ERC-Advanced grant SPLICECODE), EU-AIMS which receives support from the *Innovative Medicines Initiative* Joint Undertaking of the EU FP7, and the Kanton Basel-Stadt.

## Additional information

### Funding

| Funder | Grant reference number | Author |
|---|---|---|
| Schweizerischer Nationalfonds zur Förderung der Wissenschaftlichen Forschung | 310030B-160319 | Peter Scheiffele |
| European Research Council | SPLICECODE | Peter Scheiffele |
| The National Centre of Competence in Research | NCCR SYNAPSY | Peter Scheiffele |
| Innovative Medicines Initiatives | EU-AIMS | Peter Scheiffele |
| Boehringer Ingelheim Fonds | | Lisa Traunmüller |
| Marie-Curie Mobility Fellowship European Union | | Dietmar Schreiner |
| Werner Siemens/Opportunities in Excellence Fellowship | | Thi-Minh Nguyen |

The funders had no role in study design, data collection and interpretation, or the decision to submit the work for publication.

### Author contributions

T-MN, LT, Conception and design, Acquisition of data, Analysis and interpretation of data, Drafting or revising the article; DS, LX, Conception and design, Acquisition of data, Analysis and interpretation of data; CB, Acquisition of data, Drafting or revising the article; PS, Conception and design, Analysis and interpretation of data, Drafting or revising the article

### Author ORCIDs

Peter Scheiffele, http://orcid.org/0000-0002-9516-9399

### Ethics

Animal experimentation: All animal procedures were reviewed and approved by the Kantonales Veterinäramt Basel-Stadt (Licences 2272, 2293, 2599). The Procedures were performed in strict

accordance to the guidelines and every effort was mode to minimize suffering of the animals and to minimize animal numbers (either by replacement or optimization of procedures).

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
