## [Decision Letter]

[Editors’ note: a previous version of this study was rejected after peer review, but the authors submitted for reconsideration. The first decision letter after peer review is shown below.]

Thank you for submitting your work entitled "An alternative splicing switch shapes neurexin repertoires in principal neurons versus interneurons of the hippocampus" for consideration by *eLife*. Your article has been favorably evaluated by a Senior Editor and three reviewers, one of whom, Sacha B Nelson (Reviewer #1), is a member of our Board of Reviewing Editors. The following individual involved in review of your submission has agreed to reveal their identity: Joris De Wit (Reviewer #2).

Our decision has been reached after consultation between the reviewers. Based on these discussions and the individual reviews below, we regret to inform you that we will not be able to publish your manuscript in *eLife* in its current form.

The editorial policy at *eLife* requires that manuscripts requiring additional experimental work that would take more than approximately two months be rejected. There is also no separate "soft reject" or "reject-resubmit" category recognized. In this case, however, all three reviewers recognized the merits of the study and support publication, but also felt clarification, and importantly, additional experiments were likely needed to support the claims made in the paper. Assuming the results of these experiments are not already in hand, they would likely take several months to complete.

The Reviewing Editor has identified the following key points for revision with input from the other reviewers and Senior Editor:

1) All three reviewers felt that the absence of a synaptic deficit significantly reduced the impact of the paper and made it difficult to relate the behavioral phenotype to molecular differences between hippocampal cell types identified. Further attempts to address this are warranted. The problem is that there is no evidence tying the behavioral phenotype to the cells on which the rest of the paper focuses. The tie could be made by seeking a synaptic phenotype, but if the authors have some other way of making the link, that would be fine too.

2) Two of the reviewers felt that the relationship between cerebellin expression and neurexin function was not pursued adequately. Some additional experiments were proposed but it is also possible that this concern could be addressed with results on hand and textual clarification or by removing this section.

3) An additional concern of one reviewer was the lack of direct evidence linking Slm2 levels and AS4 splicing in these cells. While it would improve the manuscript to address this with additional manipulations this is not required for acceptance.

*Reviewer #1:*

Since the discovery that neurexins undergo extensive alternative splicing, scientists have speculated that cell type specific splicing is important for cell type specific patterns of connectivity. Several studies have provided evidence that multiple splice variants are expressed and some studies have argued that this expression is cell type specific, but this is the first study I am aware of to directly test the consequence of a cell type specific knockout of a specific isoform.

The authors first show that *Pvalb* interneurons selectively express the AS4+ isoform and that this is likely due to the fact that they lack expression of the Slm2 RNA binding protein. They also show that heterologous expression of this splice variant in culture enhances formation of inhibitory synapses. They then use a conditional allele to selectively delete this isoform in *Pvalb*+ cells in mice and show a behavioral deficit in a working memory task.

The results presented are convincing and are important because they directly test key aspects of the neurexin splice code hypothesis. Unfortunately, the paper is missing a phenotype at the synaptic level. Somatic inhibitory synapses onto pyramidal neurons appear normal and mIPSCs are not affected (nor are numbers of *Pvalb* neurons). But the findings – that the potentially relevant ligand (Cbln4) is expressed in the same *Pvalb* cells as the NrxAs4+ transcripts, and that these Nrx transcripts enhance inhibitory synapses – argue that the relevant synaptic deficit to look for are inhibitory synapses between *Pvalb* cells. At least in the neocortex, these are the dominant source of inhibition of these interneurons and disruption of this synapse might destabilize the persistent activity hypothesized to underlie working memory. Therefore, it is disappointing that the authors did not look for anatomical and physiological evidence of reduced inhibition between *Pvalb* interneurons. If the authors did look for this and did not find it, this negative result is an important one that should be included. (As an aside, SST neurons also provide some inhibitory input to *Pvalb* neurons in some forebrain structures).

I think the paper is probably important enough to publish even without this synaptic punchline, but it would have greatly increased impact with this additional data if this is feasible.

*Reviewer #2:*

Nguyen et al. investigate neurexin (*Nrxn*) isoform diversity in specific neuronal cell types and uncover a major, cell type-specific difference in *Nrxn* alternative splice site 4 (AS4) usage. They find that parvalbumin (PV)-positive interneurons exhibit a high level of AS4 inclusion (AS4+), whereas pyramidal cells are preferentially AS4-. The AS4 inclusion in α *Nrxn*s increases their ability to induce inhibitory postsynaptic specializations in cell culture. In addition, expression of AS4+ *Nrxn*s in PVs coincides with the secreted *Nrxn* ligand Cbln4, which specifically binds to AS4+ *Nrxn*s. Thus, pyramidal and PV neurons have differential molecular repertoires of *Nrxn* isoforms and *Nrxn* ligands. The authors then test the effects of deleting *Nrxn1* and *Nrxn3* AS4 in PV neurons and find no changes in PV synapse structure or function, but do find changes in short-term memory.

Overall this is a well-performed study that will be of broad interest to the field. Deciphering how cell type-specific repertoires of molecularly diverse cell surface proteins are generated and contribute to the formation of precisely wired neural circuits is an important and timely topic. I have only a few comments/suggestions that need to be addressed before publication.

1) The authors show a correlation between low PV neuron expression of the STAR-family RNA-binding protein Slm2, which regulates AS4 skipping, and high levels of AS4+ *Nrxn* isoforms in these cells. The manuscript would be strengthened by putting the causal relation between absence of Slm2 and cell type-specific exon skipping to the test. One experiment to test this would be to use AAV to misexpress Slm2 specifically in PV cells in a Cre-dependent manner and analyze whether this causes a shift towards more AS4-*Nrxn* isoforms. Another experiment would be to analyze *Nrxn* isoform repertoires in pyramidal cells of CamK2 Ribotrap mice crossed with Slm2 KO mice.

2) The interaction of Cbln4 with *Nrxn* AS4+ shown in Figure 3, and the functional relevance thereof, should be further clarified. In Figure 3, Cbln4 shows little binding to *Nrxn3α* AS4+ and Cbln1 shows some binding to *Nrxn3α* AS4+, whereas in 3F Cbln4 shows binding to *Nrxn3α* AS4+ to a similar degree as Cbln1. How can the apparent difference in Cbln binding to *Nrxn* between Figure 3 (COS cell binding vs surface retention) be explained, as both experiments measure Cbln at the surface? Does binding of Cbln4 to *Nrxn* AS4+ affect the induction of postsynaptic inhibitory specializations by AS4+ α *Nrxn*? In Figure 3, are Cbln1, which binds most strongly to *Nrxn*a AS4+, and Cbln3 also expressed in PV neurons? Finally, is expression of Cbln4 in PV neurons perhaps dependent on an *Nrxn* AS4+ repertoire?

3) The differential expression of *Nrxn* splice variants in principal vs. PV interneurons is well characterized, and the correlation of the AS4+ variant in PV cells with low Slm2 expression and high Clbn4 expression is striking. However, there is a large gap between these findings and the finding that short-term memory is impaired in the mice with PV-specific deletion of *Nrxn1/3* AS4, which limits the impact of the manuscript. The link between PV neuron-specific *Nrxn* splicing, absence of obvious synaptic abnormalities in PV synaptic boutons and the observed behavioral impairments is not clear and raises questions. Are PV neurons known to play a role in the behavioral task assayed? Does this task also rely on PV micro-circuits outside the hippocampus and do PV neurons in those circuits also display enriched expression of AS4+? In other words, is the preferential use of AS4+ in PV cells a global phenomenon or might it be brain region-specific? A further exploration of synaptic defects in PV neurons might help to bridge the current gap between PV-specific *Nrxn* isoform repertoires and behavioral defects. In addition to analyzing spontaneous synaptic transmission, the authors could analyze evoked inhibitory responses and look at plasticity of PV-pyramidal neuron synapses. This would strengthen the impact of their findings. I understand the current focus is on PV synaptic terminals, but the authors could also explore whether excitatory inputs onto PV neurons might be affected in *Nrxn1*/3*^ex21^*Pv mice.

*Reviewer #3:*

In a series of papers, Scheiffele and colleagues have analyzed alternative splicing in neurons, with particular emphasis on the neurexin and neuroligin adhesion molecules and the STAR family of splicing factors (Chih, 2006; Ijima 2011, 2014; Schreiner, 014a, 2015; Traunmuller, 2014, 2016). Here, Nguyen et al. continue this analysis, focusing on differences in α-neurexin splicing between inhibitory (PV^+^) and pyramidal (CamK2^+^) neurons in hippocampus. Using state-of-the-art methods, including RiboTrap purification of RNA from genetically identified cell types and quantitative analysis of exon usage, they show high inclusion rate of AS4 in neurexins in inhibitory relative to pyramidal neurons. Their previous work had shown that the STAR family member Slm2 leads to AS4 exclusion, and sure enough, PV^+^ cells have lower levels of Slm2 than pyramids. They then go on to assess the consequences of AS4 inclusion/exclusion in two ways. First they confirm and extend previous work showing that AS4 affects the relative activity of neurexins on inhibitory and excitatory synaptogenesis, as well as its binding to cerebellins. Second, they show that selective deletion of the AS4 exon from neurexins 1 and 3 in PV-expressing cells degrades performance of mice in an object recognition task, which is generally interpreted as indicating impairment in short-term memory.

Most previous studies on patterns and roles of alternative splicing have been performed at the level of brain regions, whereas direct evidence of cell type-specific isoform expression and function has been limited. In this regard, the work of Nguyen et al. is an important contribution to our understanding of molecular mechanisms that distinguish cell types and their functions. There are, however, some gaps in the analysis that deserve attention.

1) The authors emphasize splicing and bioactivities of *Nrxn3* in much of the text, then perform the genetic analysis only on *Nrxn1/3* double mutants. This leads to the obvious question of why *Nrxn3* single mutants weren't analyzed, or at least compared to the doubles. In fact, I think this seeming inconsistency is more a matter of style than substance. If the authors gave similar weight to the two genes in the splicing analysis, everything would be more logical. Some additional data on *Nrxn1* will be needed in the binding and bioactivity sections, unless it can be cited from previous papers.

2) It would be helpful to include data on pyramidal cells in Figure 2, for comparison with the data on interneurons.

3) The data on cerebellins is somewhat peripheral to the main point, but since the authors have chosen to include it, they need to expand it a bit. Are cerebellins believed to act in the same cells as neurexins or to be transsynaptic ligands? What does the "association" shown in Figure 3 mean? The authors seem to suggest it is not direct binding. A minor related point is that the reader may be led to believe that selective cerebellin expression is going to figure in the phenotype, whereas in fact it is just an interesting observation. This should be clarified to avoid confusion and disappointment.

4) Perhaps the most important problem is with the behavioral analysis in vivo. The authors are making the case that inclusion of *Nrxn1/3* exon 21 in hippocampal interneurons is essential for optimal hippocampal operation. Yet their cre line excises these exons from large numbers of interneurons throughout the brain and spinal cord, as well as large populations of sensory neurons and even some non-neuronal cells. Moreover, they fail to identify any physiological or morphological defects in the interneuronal synapses that could be responsible for driving the behavioral change. Thus, I'm not sure it is justifiable to relate the behavioral results in Figure 5 to the neurons analyzed in Figure 1-4.

---

## [Author Response]

[Editors’ note: the author responses to the first round of peer review follow.]

*Reviewer #1:*

*[…] The results presented are convincing and are important because they directly test key aspects of the neurexin splice code hypothesis. Unfortunately, the paper is missing a phenotype at the synaptic level. Somatic inhibitory synapses onto pyramidal neurons appear normal and mIPSCs are not affected (nor are numbers of Pvalb neurons). But the findings – that the potentially relevant ligand (Cbln4) is expressed in the same Pvalb cells as the NrxAs4+ transcripts, and that these Nrx transcripts enhance inhibitory synapses – argue that the relevant synaptic deficit to look for are inhibitory synapses between Pvalb cells. At least in the neocortex, these are the dominant source of inhibition of these interneurons and disruption of this synapse might destabilize the persistent activity hypothesized to underlie working memory. Therefore, it is disappointing that the authors did not look for anatomical and physiological evidence of reduced inhibition between Pvalb interneurons. If the authors did look for this and did not find it, this negative result is an important one that should be included. (As an aside, SST neurons also provide some inhibitory input to Pvalb neurons in some forebrain structures).*

We thank the reviewer for this suggestion. We have now quantified the density of Synaptotagmin-2 positive (PV-neuron) terminals formed onto parvalbumin-positive interneurons in control and *Nrxn1/3^ex21^* conditional knock-out mice. We did not find any significant alteration, indicating that there is not a major change in these synapses. This new data is now included in Figure 5—figure supplement 2.

*Reviewer #2:*

*[…] 1) The authors show a correlation between low PV neuron expression of the STAR-family RNA-binding protein Slm2, which regulates AS4 skipping, and high levels of AS4+ Nrxn isoforms in these cells. The manuscript would be strengthened by putting the causal relation between absence of Slm2 and cell type-specific exon skipping to the test. One experiment to test this would be to use AAV to misexpress Slm2 specifically in PV cells in a Cre-dependent manner and analyze whether this causes a shift towards more AS4- Nrxn isoforms.*

We have now performed a similar experiment to test whether Slm2 is sufficient to drive the expression of *Nrxn* AS4- isoforms when ectopically expressed. We drove expression of Slm2 in cerebellar granule cells which are Slm2-negative and primarily express *Nrxn* AS4+ variants. Ectopic expression of Slm2 was sufficient to shift splicing towards expression of AS4- variants. This data is now included in Figure 2—figure supplement 2.

*Another experiment would be to analyze Nrxn isoform repertoires in pyramidal cells of CamK2 Ribotrap mice crossed with Slm2 KO mice.*

This is a good suggestion. Unfortunately, we currently did not have this genotype combined in our mouse colony. Given that this experiment requires combination of four alleles (*CaMK2^cre^*, Rpl22-HA, and two Slm2 KO alleles) we were not able to perform this experiment in the allotted timeframe.

*2) The interaction of Cbln4 with Nrxn AS4+ shown in Figure 3, and the functional relevance thereof, should be further clarified. In Figure 3, Cbln4 shows little binding to Nrxn3α AS4+ and Cbln1 shows some binding to Nrxn3α AS4+, whereas in 3F Cbln4 shows binding to Nrxn3α AS4+ to a similar degree as Cbln1. How can the apparent difference in Cbln binding to Nrxn between Figure 3 (COS cell binding vs surface retention) be explained, as both experiments measure Cbln at the surface?*

We apologize for not having explained the difference between these two experiments appropriately in the original manuscript. Figure 3 showed experiments were Neurexin was expressed in cells and Cbln proteins expressed in a separate batch of cells and secreted into the medium were added to the Neurexin-expressing cells to assess surface binding. In a second set of assays (Figure 3) we co-expressed Cbln proteins and neurexins in the same cells (as in parvalbumin-positive interneurons both proteins are co-expressed). In this second assay configuration Cbln4 shows significantly higher interaction with Nrx3αAS4+ isoforms. This may resolve some of the conflicting data that was found in the previous literature. To clarify the two assay configurations used in our study we now included cartoons for illustration (Figure 3).

*Does binding of Cbln4 to Nrxn AS4+ affect the induction of postsynaptic inhibitory specializations by AS4+ α Nrxn? In Figure 3, are Cbln1, which binds most strongly to Nrxna AS4+, and Cbln3 also expressed in PV neurons? Finally, is expression of Cbln4 in PV neurons perhaps dependent on an Nrxn AS4+ repertoire?*

We have tested directly whether co-expression of Cbln4 together with *Nrxn*AS4+ modifies the induction of postsynaptic inhibitory specializations. We do not find a significant difference in this assay. We note that presence of endogenous Cbln4 in the neuronal culture may conceal a function for Cbln4. Loss of function studies for Cbln4 will be required to resolve this. The new results are now shown in Figure 3 and Figure 3—figure supplement 1.

Regarding the expression of Cbln1 and 3 in PV^+^ cells we note that both transcripts show only very low expression in hippocampus when probed with qPCR assays. The same assays show robust detection in the cerebellum (new data in Figure 3—figure supplement 1). Thus, we interpret our observation that Cbln1 and 3 transcripts are poorly or not at all detected in hippocampal PV^+^ cells as an indication that they are not significantly expressed and are unlikely to play a major role in these cells. This conclusion is consistent with previous in situ hybridization data in the literature (Miura et al., European Journal of Neuroscience, 2006).

3) The differential expression of Nrxn splice variants in principal vs. PV interneurons is well characterized, and the correlation of the AS4+ variant in PV cells with low Slm2 expression and high Clbn4 expression is striking. However, there is a large gap between these findings and the finding that short-term memory is impaired in the mice with PV-specific deletion of Nrxn1/3 AS4, which limits the impact of the manuscript. The link between PV neuron-specific Nrxn splicing, absence of obvious synaptic abnormalities in PV synaptic boutons and the observed behavioral impairments is not clear and raises questions. Are PV neurons known to play a role in the behavioral task assayed? Does this task also rely on PV micro-circuits outside the hippocampus?

The object recognition task is known to rely on the hippocampus but also the entorhinal cortex (Cohen & Stackman, Behav. Brain Res., 2015). Donato et al. (Nature, 2013) demonstrated for area CA3 of the hippocampus that PV^+^ neurons activation decreases whereas PV-neuron inhibition increases performance of mice in the novel object recognition task.

*And do PV neurons in those circuits also display enriched expression of AS4+? In other words, is the preferential use of AS4+ in PV cells a global phenomenon or might it be brain region-specific?*

To further explore splicing regulation at AS4 we performed RiboTrap purifications from neocortical neurons. In our splicing analysis we find that – similar to the hippocampus – CamK2-positive neurons preferentially express AS4- whereas parvalbumin-positive interneurons preferentially express AS4+ (particularly for *Nrxn3*). Thus, the preferential exon use reported for the hippocampus is shared in neocortical cells. This new data is included in Figure 2—figure supplement 1.

*A further exploration of synaptic defects in PV neurons might help to bridge the current gap between PV-specific Nrxn isoform repertoires and behavioral defects. In addition to analyzing spontaneous synaptic transmission, the authors could analyze evoked inhibitory responses and look at plasticity of PV-pyramidal neuron synapses. This would strengthen the impact of their findings. I understand the current focus is on PV synaptic terminals, but the authors could also explore whether excitatory inputs onto PV neurons might be affected in Nrxn1/3^ex21^Pv mice.*

We appreciate that identifying a synaptic alteration that might be responsible or at least correlate with the behavioral phenotype would greatly strengthen this work. We have initiated some paired recordings in the knock-out mice, however, these experiments are very time-consuming and have not yet yielded novel insights.

To explore further whether the behavioral alterations coincide with alterations in neuronal network activity in the hippocampus, we compared c-fos-immunoreactivity in control and *Nrxn1/3^ex21^* conditional knock-out mice. We observed a significant increase in the density of c-fos^+^ cells in area CA3 of the mutant mice as well as a trend towards increased density of c-fos^+^ cells in the dentate gyrus and area CA1. This indicates that parvalbumin-neuron–specific ablation of the AS4+ insertions results in an overall increase in network activity in the hippocampus. These new results are now presented in Figure 6.

*Reviewer #3:*

*[…] 1) The authors emphasize splicing and bioactivities of Nrxn3 in much of the text, then perform the genetic analysis only on Nrxn1/3 double mutants. This leads to the obvious question of why Nrxn3 single mutants weren't analyzed, or at least compared to the doubles. In fact, I think this seeming inconsistency is more a matter of style than substance. If the authors gave similar weight to the two genes in the splicing analysis, everything would be more logical.*

The reason that we did emphasize *Nrxn3* is that *Nrxn3* transcripts were most strongly enriched in the parvalbumin-RiboTrap preparations. However, clearly *Nrxn1* and *2* are also expressed in these cell preparations. We now included data on the *Nrxn3* single mutants, including weight and Mendelian frequency (Figure 4—figure supplement 1), anatomical analysis (Figure 5—figure supplement 1) and data from mIPSC recordings (Figure 5—figure supplement 1). We do not detect any significant changes in these single mutants.

*Some additional data on Nrxn1 will be needed in the binding and bioactivity sections, unless it can be cited from previous papers.*

The manuscript did contain data on the Cbln4-Nrx1 interactions. We apologize if this was not sufficiently visible in the previous submission. This previous data is presented in Figure 3. In addition, we now included data on induction of gephyrin-positive structures by Nrx1α isoforms in presence and absence of Cbln4 (Figure 3—figure supplement 3D and 3E). For both, Nrx1α and Nrx3α we do not observe a significant alteration in the synaptogenic activity with co-expression of Cbln4.

*2) It would be helpful to include data on pyramidal cells in Figure 2, for comparison with the data on interneurons.*

We have now quantified the percentage of cells marked in the CamK2^Ribo^ mice that express Slm2. We find that Slm2 is expressed in >90% of CamK2^Ribo^-positive cells (Figure 2).

*3) The data on cerebellins is somewhat peripheral to the main point, but since the authors have chosen to include it, they need to expand it a bit. Are cerebellins believed to act in the same cells as neurexins or to be transsynaptic ligands?*

Our expression data demonstrates that Cbln4 is significantly enriched in PV^+^cells as compared to Camk2-positive cells. This means, *Nrxn* AS4+ isoforms and Cbln4 are co-expressed and indicates that they are transported to the PV-cell surface and then might engage postsynaptic ligands. This model is analogous to the action of *Nrxn*-Cbln1 complexes in the mouse cerebellum.

*What does the "association" shown in Figure 3 mean? The authors seem to suggest it is not direct binding. A minor related point is that the reader may be led to believe that selective cerebellin expression is going to figure in the phenotype, whereas in fact it is just an interesting observation. This should be clarified to avoid confusion and disappointment.*

The two assays presented in Figure 3 are cell binding assays. While the interactions that we see are consistent with direct *Nrxn*-Cbln interactions we cannot completely exclude that additional co-factors contribute to the interaction. We have now clarified this in the text (subsection “AS4 + splice insertions selectively enhance the function of neurexins towards GABAergic postsynaptic components”, first paragraph). We have also clarified that it remains to be shown whether Cbln4 mediates any of the functions of *Nrxn* AS4+ isoforms in PV^+^ cells.

*4) Perhaps the most important problem is with the behavioral analysis* in vivo*. The authors are making the case that inclusion of Nrxn1/3 exon 21 in hippocampal interneurons is essential for optimal hippocampal operation. Yet their cre line excises these exons from large numbers of interneurons throughout the brain and spinal cord, as well as large populations of sensory neurons and even some non-neuronal cells. Moreover, they fail to identify any physiological or morphological defects in the interneuronal synapses that could be responsible for driving the behavioral change. Thus, I'm not sure it is justifiable to relate the behavioral results in Figure 5 to the neurons analyzed in Figure 1-4.*

We agree with the reviewer that it is unclear whether the conditional ablation of *Nrxn1/3* AS4 insertions in PV^cre^-positive cells modifies the behavior due to an alteration in hippocampal interneurons or other cells that are targeted by the *PV^cre^* line. The object recognition test relies in part on normal hippocampal function and performance of mice in this test has been linked to the activity of PV^+^-cells in the hippocampus (Donato et al., Nature, 2013). To assess whether hippocampal network activity might be altered in *Nrxn1/3* conditional knock-out mice we measured the density of c-fos^+^ cells in the hippocampus. We find that the density of c-fos^+^ cells is elevated in the mutant mice as compared to littermate controls (Figure 6). Thus, PV-cell-specific deletion of exon21 in *Nrxn1* and *3* indeed results in a modified hippocampal network. Whether this modification is causal for the altered behavior of the mice remains to be explored. We have now extended the Discussion regarding this limitation of our study.